# Modelling Flood Losses to Microbusinesses in Ho Chi Minh City, Vietnam

Anna Buch[1,2,3], Dominik Paprotny[3], Kasra Rafiezadeh Shahi[1,4], Heidi Kreibich[1], Nivedita Sairam[1]

[1]German Research Centre for Geosciences (GFZ), Section 4.4 Hydrology, Potsdam, 14473, Germany

[2]Heidelberg University, GIScience Research Group, Heidelberg, 69117, Germany

[3]Potsdam Institute for Climate Impact Research (PIK), Transformation Pathways, Potsdam, 14473, Germany

[4]Potsdam Institute for Climate Impact Research (PIK), Earth System Analysis, Potsdam, 14473, Germany

*Correspondence to*: Anna Buch (anna.buch@uni-heidelberg.de), Nivedita Sairam (nivedita.sairam@gfz.de)

**Abstract.** Microbusinesses are important sources of livelihood for low- and middle-income households. In Ho Chi Minh City (HCMC), Vietnam, many microbusinesses are set up on the ground floor of residential houses susceptible to urban floods. Increasing flood risk in HCMC threatens the financial resources of microbusinesses by damaging business contents and causing business interruption. Since flood loss estimations are rarely conducted at object-level resolution and are often focused on households or large companies, the commercial losses suffered by microbusinesses are often overlooked. This study aims to

derive the drivers of flood losses [%] in microbusinesses by applying a Conditional Random Forest to survey data (relative business content losses: n=317; relative business interruption losses: n=361) collected from microbusinesses in HCMC. The variability of the losses to business content and due to business interruption were adequately explained by the revenues of the businesses from monthly sales, age of the building where the business is established, and hydrological characteristics of the flood. Based on the identified drivers, probabilistic loss models (non-parametric Bayesian Networks) were developed using a

combination of data-driven and expert-based model formulation. The models estimated the flood losses for HCMC's microbusinesses with a mean absolute error of 3.8 % for content losses (observed mean: 4.7 %) and 18.7 % for business interruption losses (observed mean: 18.2 %). The Bayesian Network model for business interruption performed with a similar predictive performance when it was regionally transferred and applied to comparable survey data from another Vietnamese city, Can Tho. The flood loss models introduced in this study make it possible to derive flood risk metrics specific to

microbusinesses to support adaptation decision making and risk transfer mechanisms.

**Plain Language Summary.** Many households in Vietnam depend on revenues from microbusinesses (shophouses). However, losses caused by regular flooding to the microbusinesses are not modelled. Business turnover, building age and water depth are found to be the main drivers of flood losses to microbusinesses. We built and validated probabilistic models (non-

parametric Bayesian Networks) that estimate flood losses to microbusinesses. The results help in flood risk management and adaption decision making for microbusinesses.

## 1 Introduction

Comprehensive risk management requires empirical evidence on drivers of risk and assessment of potential impacts. The lack of information on vulnerability of certain economic sectors or social groups, and their often limited participation in local risk

management, in turn foster a lack of awareness among decision-makers leading to biased risk management strategies. As impacts of climate change become more severe, comprehensive risk management that protects society as a whole is imperative - in particular the vulnerable and under-represented groups. However, it is often not feasible in low- and middle-income countries due to poor data availability. An example of a vulnerable economic sector in a society with a high flood risk, explored in this study, are microbusinesses in Ho Chi Minh City (HCMC), Vietnam. These micro-sized companies are quite common

across Southeast (SE) and South (S)-Asia; a description of their operations and economic relevance in regard to Vietnam's urban areas are provided in Sect. 2.1.

In addition to the largely studied structural damages, the commercial sector, specifically, microbusinesses also suffer directly from economic loss of business content (e.g. inventory, goods, equipment) and due to business interruption. The latter refers to the decline in business revenues due to interrupted operations of flood-affected businesses during a reference period such

as the flood month or period of flooding (Meyer et al., 2013; Chinh et al., 2016). However, our definition of interruption losses does not consider long-term losses or impacts on businesses outside the flood zone. The literature on commercial losses often focuses on companies of various sizes in Europe or the US and these studies indicate that indirect losses represent a significant share of flood consequences (e.g. Hallegatte, 2008; Merz et al., 2010; Koks and Thissen, 2016; Sieg et al., 2019; Tsinda et al., 2019). Since the business structures and available resources for larger firms differ considerably from those of small- and micro-

sized companies (Leitold and Diez, 2019), the state-of-the-art approaches for commercial flood loss modelling are not generalizable to Vietnam's microbusinesses. However, the better the drivers of flood losses for a specific sector are understood, the more informed loss assessments can be made and investments towards flood adaptation improved (Sieg et al., 2017).

Modelling flood losses in low- and middle-income countries is often hampered by the lack of comprehensive and open-source data, which necessitates reliance on primary data collection campaign. The lack of information on flood losses among

microbusinesses is explained by the fact that they mainly operate in the informal sector, which makes it difficult to record and thus to estimate their flood losses (Garschagen, 2015; Rand and Tarp, 2020). Despite these limitations, some studies have analysed and modelled content losses to micro-, small- and medium-sized companies in SE and S-Asia (Chinh et al., 2016; Wijayanti et al., 2017; Samantha, 2018). To the authors knowledge, there is no existing analysis elucidating the drivers of flood losses specific to microbusinesses in the context of low- and middle-income countries. However, the identification of

the loss drivers is crucial to develop meaningful flood loss models that capture the role of the drivers in influencing losses (Rözer et al., 2019). The heterogeneity in flood loss processes at the object-level necessitates the development of multi-

variable, probabilistic approaches capable of capturing non-linear effects (Schröter et al., 2014; Vogel et al., 2014; Rözer et al., 2019; Paprotny et al., 2020; Paprotny et al., 2021; Rafiezadeh Shahi et al., 2024). The absence of such probabilistic loss models in the context of microbusinesses impedes quantification and inclusion of uncertainties for adaptation decision making.

Furthermore, multivariate flood loss models are rarely evaluated under conditions other than those under which they were developed, consequently their applicability for spatial/temporal transfers remains unknown (Apel et al., 2009; Gerl et al., 2014; Ootegem et al., 2017; Vogel et al., 2018; Amadio et al., 2019). Our study aims to address these limitations in the state-of-the-art flood loss modelling approaches for microbusinesses in the context of low- and middle-income countries by deriving empirical evidence on the drivers of flood losses to microbusinesses in HCMC; calibrating and validating process-based

Bayesian Network (BN) models for HCMC that predict content and business interruption losses; and evaluating the transferability of the models by applying them on comparable data from a different city (Can Tho).

The manuscript is organized as follows (see, Supplementary Information Fig. S1): Section 2 comprises an overview of the research domain and the empirical survey datasets used in the study, Sect. 3 the methodology implemented including feature selection and the development of probabilistic flood loss models, Sect. 4 presents and discusses the results of this study,

followed by conclusions in Sect. 5.

## 2 Case studies and data

### 2.1 Case studies

HCMC is one of the world's most exposed cities to flood risk under current and future conditions (Hallegatte et al., 2013; Scussolini et al., 2017). Similar to other SE-Asian metropolises, HCMC lies in a river delta area close to the coast. These

densely populated, flat, riverine and coastal regions experience regular flooding in particular during the rainy season (Garschagen, 2015; Tierolf et al., 2021; Nguyen et al., 2021). In HCMC, these regular floods are often the result of compound events caused by the simultaneous occurrence of high tides, heavy rainfall and high flows of the Saigon and Dong Nai rivers and their tributaries (Tran, 2014; Thuy et al., 2019). Other large cities in the delta areas of South Vietnam also experience regular urban flooding, for instance, Can Tho City in the Mekong Delta (abbreviated as Can Tho). Urban floods in Can Tho

are predominantly fluvial in nature, such as a major flood event in 2011. Despite the ongoing efforts to improve protection and adaptation measures on private and municipal levels, climate change and the ongoing growth of these important economic centres increase their risk to urban flooding (Güneralp et al., 2015; Rentschler et al., 2022). The existing infrastructure and adaptation measures in these cities are unable to counterbalance the new risks caused by intensified flood events and ongoing urban pressure (e.g. Bouwer, 2011; Bloch et al., 2012; Formetta and Feyen, 2019; Kreibich et al., 2022). The overview map in

Fig. 1 illustrates the locations of both cities (HCMC, Can Tho) used in the case studies of this paper.

We define microbusinesses, including household-businesses, according to the definition of the World Bank, as very small businesses with less than ten employees. However, this general definition for microbusinesses needs to be adapted to the regional context of SE and S-Asia. Microbusinesses in these countries tend to employ usually less than three people. In most

cases, microbusinesses are located on the ground floor of a building with residences on the upper floors, commonly called
shophouses in Vietnamese cities. Microbusinesses provide an important source of income for unemployed family members
and people with limited opportunities on the labour market, likewise migrant workers and people who received less possibilities
of schooling (Samantha, 2018). Together with the operations of small and medium-sized companies (SMEs), microbusinesses
drive the rapid economic developments of many SE-Asian states since the last decades (Trinh and Thanh, 2017). According
to Vietnam's economic census of 2017, it is estimated that around 75 % of all enterprises are microbusinesses in the country
(General Statistics Office, 2018). Vietnam's microbusinesses engage around 11 % of all employees (General Statistics Office,
2018) and the density of microbusinesses is particularly high in economic centres such as HCMC and other delta-cities like
Can Tho. The economic importance of HCMC becomes evident when the region's contribution to Vietnam's total economic
output is considered - the HCMC region accounts for approximately 40 % of the national GDP (General Statistics Office,
2018). These values highlight the relevance of microbusinesses for Vietnam's local and national economy.

The microbusiness owners are particularly vulnerable to the negative consequences of regular flooding due to their limited
financial resources and inadequate support by local authorities and the government (Leitold et al., 2020). As a consequence,
the owners often rely on their neighbouring network to cope with flooding (Chinh et al., 2016; Chinh et al., 2017; Leitold and
Revilla Diez, 2019; Leitold et al., 2021). Bank loans or microcredits are less common due to the usually rather low credit rating
(Patankar, 2019). In terms of flood losses this means that repair measures and other business investments are often directly
financed by the savings of the microbusiness owner. Insufficient or missing flood insurance policies can further exacerbate the
situation of flood-affected businesses (KPMG, 2016; Patankar, 2019). Besides temporal decline in revenues, repair costs or
poor future prospects, worse case impacts may include business closures or unemployment among business owners and their
employees (Bloch et al., 2012).

**2.2 Data – post-flood survey of microbusinesses**

The flood loss models for microbusinesses are built using empirical data from HCMC and the transferability of the models is
evaluated using empirical data from Can Tho. Both datasets are based on in-person structured surveys undertaken with flood-
affected microbusinesses. The owner or the manager of the microbusiness was asked to respond to the survey. They were
informed about the project, how their responses would be used, and that they could leave the survey at any time. No personal
or health-related information was collected in either survey. The data is stored and handled exclusively within the German
Research Centre for Geosciences (GFZ) in compliance with data privacy and data protection regulations.
The workflow of this study presents in the first main step the preparation of the survey datasets (see, Supplementary
Information Fig. S1); the respective key aspects are described in detail in the following subsections and in the Supplementary
Information, Sect. S1.

### 2.2.1 Ho Chi Minh City

The survey at HCMC was conducted during September-October 2020 and collected responses of 250 microbusinesses which experienced flooding between 2010 and the time of the survey (2020). The majority of microbusinesses surveyed in HCMC are shops or retailers (76 %) mostly selling groceries or other everyday objects. Around 17 % are services, such as restaurants or for reparations, and only 7 % produce consumer goods or processes raw materials. The presented shares of the business sectors in the HCMC survey are representative for entire Vietnam (General Statistics Office, 2018).

In order to achieve a reasonable representation of HCMC, we selected the districts with the most frequent flood risk and heterogeneity in socio-economic conditions. Within each district, the shophouses were chosen randomly. The sample size in each district was not chosen based on statistical considerations, but on recommendation from local experts.

The interviewees could respond to questions on two flood events – the most severe and the most recent event. However, not every interviewee provided information for both events, which leads to a number of 397 loss records in the HCMC dataset.

Each record in this dataset comprises information about one or two types of flood losses experienced during an event. In detail, 361 samples of the loss records contain information about business interruption losses reported as relative values (e.g. reduced sales and production), while a similar sample size comprises flood losses to business content but reported as monetary values (e.g. to furniture, electrical devices, stored products and vehicles). The conversion of the latter loss type to relative scales reduced it to 317 samples (relative content loss) by using the value of business content as exposure information (see,

Supplementary Information Sect. S1). Consequently, the sample size referring to relative content losses (n=317) is smaller than for relative interruption losses (n=361). Hereinafter both types of relative flood losses are referred to as flood loss variables (Table 1).

Figure 1 visualizes the approximate locations of the microbusinesses surveyed in HCMC. However, their exact geolocations are not shown to protect the anonymity of the interviewees. Furthermore, the map shows that the surveyed microbusinesses

are often located near to an open channel or tributary river.

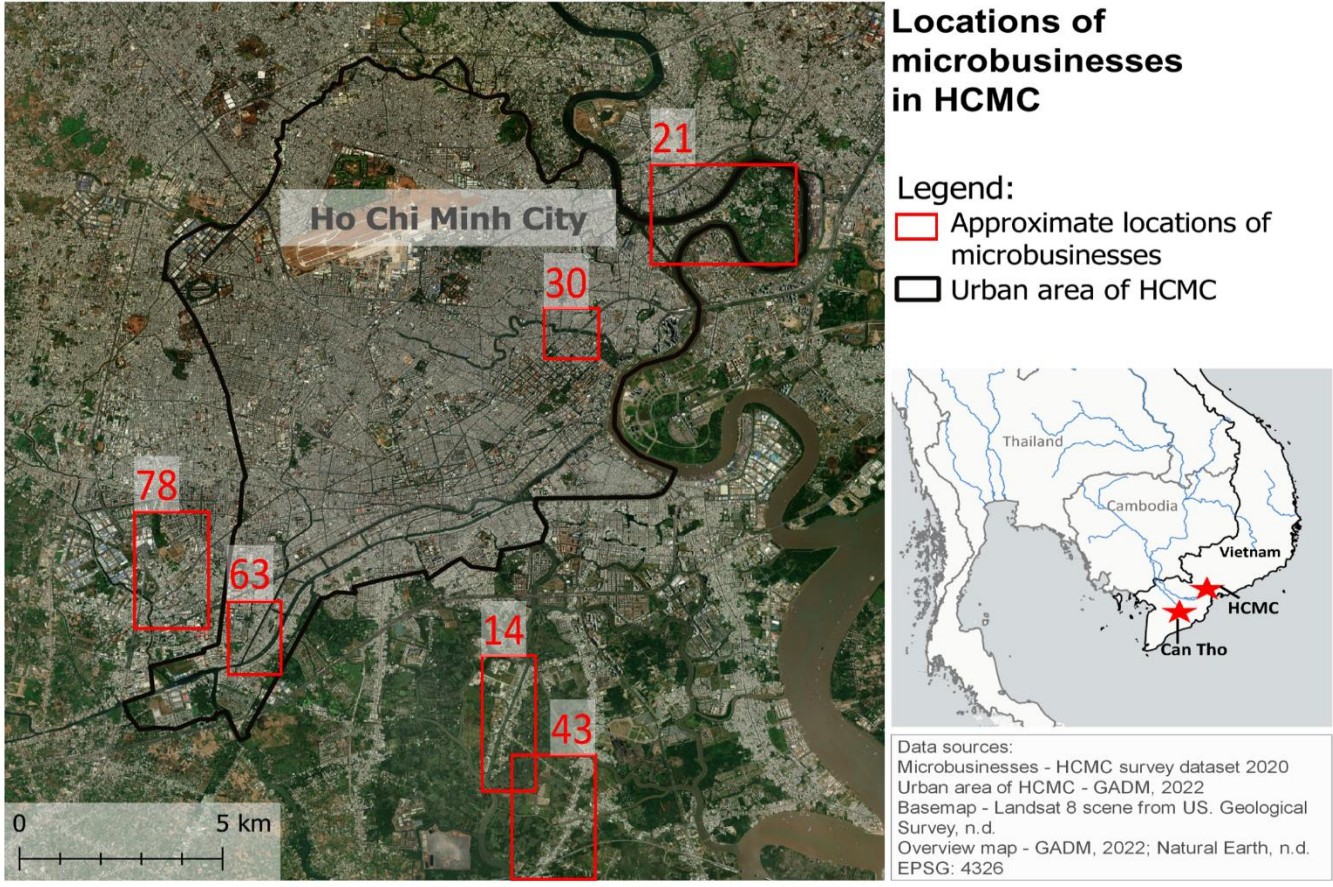

**Figure 1: Approximate locations of surveyed microbusinesses affected by floods between 2010 and 2020 in Ho Chi Minh City (HCMC) marked by red squares; the exact geolocations are not shown to protect the interviewees' anonymity. The values above the squares refer to the number of microbusinesses located in each square. The geolocation of one microbusiness was highly inaccurate and thus it is not shown in the map. The overview map on the lower right side shows the locations of both study areas (HCMC and Can Tho) in Southeast Asia. However, the geolocations of microbusinesses in Can Tho were not reported.**

### 2.2.2 Can Tho

Between August and December 2011, severe flooding affected several districts of Can Tho, causing damage to various economic sectors. The survey was undertaken in January-February 2012 and received responses from 373 microbusinesses out of which 313 furnished information on losses to business content and due to business interruption. The questionnaire is comparable to the survey undertaken in HCMC, except that each interviewee reported only about the most severe flood event during 2011 and provided information about the value of their business content. The latter was used to calculate relative content losses. Furthermore, the microbusinesses' locations were not queried. All other pre-processing steps were the same as for the HCMC data.

The value distributions of common variables from the HCMC and Can Tho survey are shown in the Supplementary

Information, Fig. S2. Compared to the HCMC survey, the Can Tho dataset includes fewer microbusinesses operating in the

trading sector (46 %). Consequently, more respondents provide services (45 %) or belong to Can Tho's manufacturing sector

(9 %). Details on the pre-processing of the Can Tho survey data are provided in the Supplementary Information, Sect. S1.

In order to derive the drivers of flood losses and to develop the loss models, the 14 pre-processed candidate predictors from

the HCMC dataset are used (Table 1), while the data from Can Tho is used to assess the models' transferability.


**Table 1: Candidate predictors and target variables of flood losses from HCMC**

| Candidate predictor | Abbreviation | Value range [Mean, Median] | Explanation |
|---|---|---|---|
| Water depth [cm] | water depth | 1–150 [34; 30] | Water depth refers to the measured flood water level above the ground floor of the shophouse. |
| Inundation duration [h] | inundation duration | 0.2–240 [11; 3] | Duration of flood inundation of the shophouse. |
| Contamination (indicator) | contamination | 0: no visible 1: light 2: heavier [x; 1] | Type of visible contamination of the flood water. |
| Flow velocity [m/s] | flow velocity | 0.1–0.5 (calm – turbulent) [x; 0.3] | Flow velocity of flood water on the street. |
| Structural precautionary measures (indicator) | structural measures | 0.0–1.0 [0.2; 0.0] | Ratio between number of implemented measures and number of possible measures. These measures are often implemented during major renovations or building constructions. They comprise the usage of water-resistant building material and the elevation of the building or parts of it. |
| Non-structural precautionary measures (indicator) | non-structural measures | 0.0–1.0 [0.4; 0.3] | Ratio of number of implemented measures and number of possible measures. These measures need to be purchased before the flood event. They are quite affordable compared to structural measures and comprise wet-proofing of valuables, installation of the electricity control system at a higher level, acquisition of mobile water-barriers and pumping equipment. |
| Emergency measures (indicator) | emergency measures | 0.0–1.0 [0.4; 0.5] | Ratio of number of implemented measures and number of possible measures. These measures can be applied shortly before or during the flood event. They comprise saving of documents, relocation of furniture, vehicles or products., usage of sandbags and sealing of doors and windows. |
| Building age [years] | building age | 0–100 [20; 18] | The age of the shophouses at the time of flooding. |
| Building area [sqm] | building area | 12–850 [87; 74] | Building footprint of the shophouse. |
| Flood experience [n] | flood experience | 3–151 [82; 76] | Number of experienced floodings between 2010 and 2020. |
| Flood resilience (indicator) | resilience | 0–5 (weak – strong) [x; 3] | Interviewee's appraisal of support by authorities or the neighbourhood. |
| Number of employees [n] | no. employees | 1–9 | Number of employees. |

| | | [x; 2] | |
|---|---|---|---|
| Average monthly income [Euro 2020] | mthly. income | 18–3314 [430; 295] | Available monthly income of the interviewee, in most cases the owner of the microbusiness. |
| Average monthly sale [Euro 2020] | mthly. sales | 92–2762 [370; 276] | Averaged revenue from monthly sales and production. The variable is representative for the value and quantity of goods and products hold by an individual microbusiness, i.e. it reflects the business size and type. |
| **Flood loss variables** | | | |
| Relative business interruption loss [%] | rbred | 0–100 [18.2; 10] | Decline in revenues due interrupted business operations (e.g. reduced production and sales) during the flood event [%]. The decline is relative to the potential revenue that would be generated without the flood. The values were reported as integers: 0 % represents no business interruption; 100 % a complete business downtime during the flood event. |
| Relative content loss [%] | rcloss | 0.0–93.5 [4.7; 0.0] | Relative loss to business content (e.g. machinery, goods and products) split into chance and degree of content loss. Values equal to 0.0 % represent no content losses (zero-loss case), values > 0.0 % occurred flood losses relative to the value of business content. |
| Chance of content loss | chance of rcloss | 0, 1 (zero-loss, loss) [x; 0] | Chance of flood losses to business content. 0 represents the absence of content loss (zero-loss case); 1 the occurrence of content loss (loss case). |
| Degree of content loss [%] | degree of rcloss | 0.2–93.5 [12.3; 4.0] | Flood losses relative to the value of business content, only loss cases [%]; Values close to 0 % represent minor flood losses to business content; 100 % the entire loss of business content. |

## 3 Methodology

Our approach for modelling flood impacts specific to microbusinesses consists of two components (see, Supplementary Information Fig. S1). First, we identify the drivers of content and interruption losses to HCMC's microbusinesses based on the set of candidate predictors (Table 1). For this feature selection, a variant of Random Forest – Conditional Random Forest – was chosen since it provides a feature importance method not biased towards correlated predictors (see, Sect. 3.1). Second, we calibrate probabilistic loss models – non-parametric Bayesian Networks – specific to microbusinesses based on the identified drivers (see, Sect. 3.2).

Since more than half of the businesses in both cities reported no or only marginal content losses (see, Supplementary Information Fig. S4.a and Fig. S4.b), we model the chance of loss to business content separately from the degree of loss. The former represents the absence or presence of content loss to microbusinesses and is binary (absence/presence), while the latter represents the severity of experienced loss and is a continuous value (0, 100). In contrast, a majority of the businesses reported

occurred interruption losses, hence the aspects of business interruption loss (chance and degree of interruption loss) were not considered separately (see, Supplementary Information Fig. S3).

The predictive performances of the Machine Learning (ML)-model used for feature selection and the flood loss models were assessed by the Mean Absolute Error (MAE), Root Mean Square Error (RMSE), Mean Bias Error (MBE) and Symmetric Mean Absolute Percentage Error (SMAPE). The MAE metric is chosen due to its outlier robustness as a selection criterion for the cross-validation of the ML-based models (Chicco et al., 2021). The equations of the performance metrics are listed in the Supplementary Information, Table S1.

### 3.1 Feature selection

### 3.1.1 Conditional Random Forest

The candidate predictors for flood losses presented in Table 1 exhibit a moderate to high degree of multicollinearity, for instance, the flood-related features are strongly correlated to each other. For this reason, Conditional Inference Trees were applied to account for these correlations during feature selection. Conditional Inference Trees were initially introduced by Hothorn et al. (2006) and extended by Strobl et al. (2007) to an ensemble of trees, a so-called Conditional Inference Random Forest (CRF). Each tree is grown only by a subset of features, which were identified before as significant based on their p-values (Hothorn et al., 2006). By this approach predictive features are identified, despite their potential collinearity to other candidate predictors. The choice of an unbiased version of the permutation-based feature importance method – namely Conditional Permutation Importance (CPI) – further reduces the chance of biased importance scores for correlated features (Debeer and Strobl, 2020). The CPI accounts on linear and non-linear interactions of correlated predictors using a chi-square test (Debeer and Strobl, 2020). Though the CPI is a measure well suited for the feature selection from CRF models (Levshina, 2020), the method is rather computationally expensive, but applicable for the presented approach due to the rather small sets of training samples.

For each of the flood loss variables a CRF model was trained and evaluated via nested cross-validation. Nested cross-validation is a state-of-the-art technique for determining an unbiased generalisation ability of a model (Krstajic et al., 2014). It is particularly recommended for relatively small datasets (Brill, 2022; Liu et al., 2022). Repeated 10 inner folds were used for hyperparameter tuning and 10 outer folds for performance evaluation of the estimators. Of these 10 evaluated estimators, the estimator with the best performance (smallest MAE-score) was used for feature selection, i.e. for identifying the drivers for the degree of content loss and relative interruption loss to microbusinesses.

### 3.2 Probabilistic flood loss models for microbusinesses

### 3.2.1 Probabilistic logistic regression

The chance of content loss, as one component of relative content loss, is modeled using a probabilistic logistic regression model, applied on the candidate predictors from Table 1. To prevent model overfitting, probabilistic logistic regression

incorporates L1 and L2 regularization, which effectively manage multicollinearity in the feature space. The model returns the probability of assigning a microbusiness to either zero-loss or loss category. However, the sample sizes between both categories are imbalanced (see, Supplementary Information Fig. S4.a). To overcome this imbalance, the logistic regression model was trained on a weighted sample of zero-loss and loss cases. Similar to the CRF (see, Sect. 3.1.1), the logistic regression model was also trained and evaluated by nested cross-validation consisting of 10 inner and 10 outer folds. However, we used all validated classifiers for modelling the chance of content loss rather than a single classifier due to their moderate predictive performance.

### 3.2.2 Bayesian Network

Bayesian Networks (BNs) are probabilistic, graphical models with many applications to flood loss modelling (Vogel et al., 2014; Wagenaar et al., 2018; Rözer et al., 2019; Paprotny et al., 2020; Paprotny et al., 2021; Rafiezadeh Shahi et al., 2024). They perform better in regional transfer settings compared to other ML-based models, such as regularized linear regressions, since BNs can be applied on incomplete information. Furthermore, they have the benefit of explicitly representing the dependency structures, quantifying uncertainty and the possibility of including expert knowledge alongside data. In more detail, the dependency structure of a BN represents (assumed) causal relations between variables, these dependencies can be set based on knowledge or logical conclusions.

In this study, non-parametric Bayesian Networks, were chosen for modelling the degree of content loss and for modelling the relative business interruption loss. As the term "non-parametric" indicates, this type of Bayesian Network does not rely on prior assumptions about the distribution of the data (Du and Swamy, 2019). Non-parametric BNs were first introduced by Kurowicka and Cooke (2006) and later extended by Hanea et al. (2006; 2015). They rather make use of the ranks of the empirical data which is favourable in terms of the varying distributions of flood losses and their potential drivers. These drivers are used to construct the graphs of the BNs. Confirmed by the Cramer-von Mises measure for the single variable-pairs of the BN graphs, the joint distributions of the variables are represented by Gaussian copulas.

The constructed flood loss models are calibrated and validated on the flood losses reported in HCMC. The performance of the single Bayesian Network model for relative interruption loss was determined by 5-fold cross-validation, while the performance of the modelling approach used for relative content loss was assessed by calculating the prediction bias directly between the reported losses and their probabilistic estimates.

The transferability of these models is assessed based on their performance in predicting flood losses in Can Tho. The performance of the models for each prediction task is benchmarked against the performance of a reference Random Forest (RF) model (Chinh et al., 2017).

## 4. Results and discussion

The section is structured as follows. Firstly, the performance metrics of the CRF model are reported and the most important flood loss drivers for microbusinesses are derived and discussed briefly (see, Sect. 4.1). Subsequently, the identified drivers are used to construct the Bayesian Network flood loss models (see, Sect. 4.2). The loss models are validated (see, Sect. 4.3) and their transferability to other delta-cities is tested using the survey data from Can Tho as a case study (see, Sect. 4.4). Finally, the model uncertainties and the limitations of the proposed approach are discussed (see, Sect. 4.5).

**4.1 Drivers of flood losses to microbusinesses**

The cross-validation of the CRF model shows that all its estimators, validated on the outer folds of the nested cross-validation, have similar moderate performances in predicting the degree of content losses and the relative interruption losses. Furthermore, the similar sets of hyperparameter values across the validated estimators show that the applied ML-algorithm is suitable for both prediction tasks. The prediction of the degree of content losses results in an averaged MAE of 12.8 %, RMSE of 18.4 %,
MBE of -0.2 % and SMAPE of 51.4 %, while the prediction of relative interruption losses leads to an averaged MAE of 17.5 %, RMSE of 22.6 %, MBE of 0.3 % and SMAPE of 59.9 %. However, high SMAPE scores are caused by less severe cases of content loss being overestimated, while moderate and severe loss cases are often underestimated by the estimators. The same applies to the prediction of interruption losses.

Revenues returned from business operations (mthly. sales) are the most influencing factor for the severity (degree) of loss to
260 business content, while the number of applied emergency measures has the greatest impact on interruption losses. Further main drivers for the degree of content loss and relative interruption loss are the age of the shophouse (building age), hydrological variables and the monthly income (Fig. 2.a and Fig. 2.b).

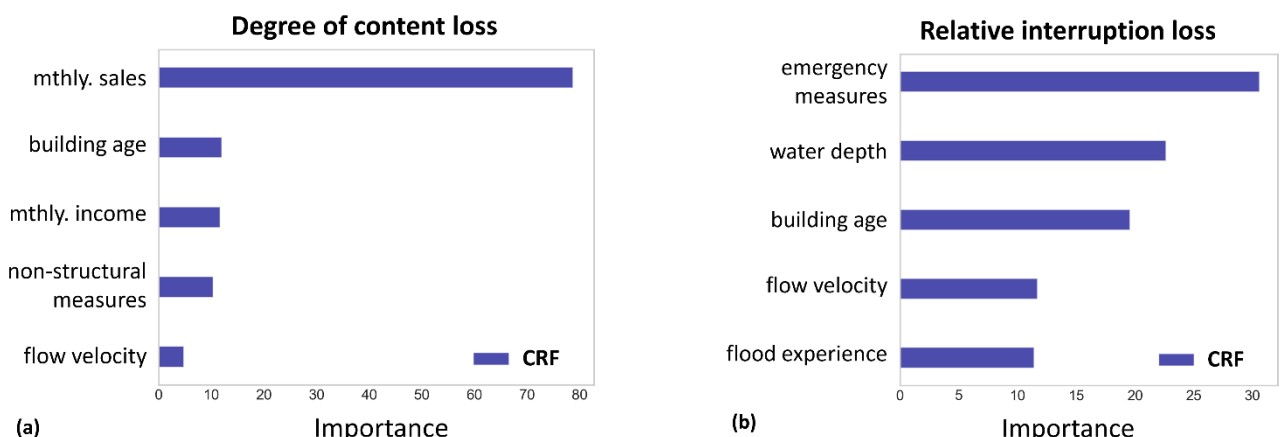

**Figure 2: Feature importance of the best-performed Conditional Random Forest (CRF) estimator for predicting (a) the degree of**
265 **losses to business content and (b) for relative losses due to business interruption to microbusinesses in HCMC. Only the five most predictive features are shown.**

The identified drivers of flood losses to microbusinesses in HCMC differ partly from those of less flood-experienced companies in high-income countries. For instance, in Germany, the company's flood experience, size (number of employees) and the building area were identified as relevant for larger companies (e.g. Kreibich et al., 2007 (flood experience); Sieg et al., 2017 (employees - content loss); Sultana et al., 2018 (employees - interruption loss); Schoppa et al., 2020 (building area)). However, these factors were not found to be critical in the case of HCMC. Of theses factors, the missing role of flood experience could be explained by HCMC's regular floodings which lead to a high level of adaptive behaviour across the residents (Harish et al., 2023).

## 4.2 Bayesian Network flood loss models

The graph of the non-parametric Bayesian Network for estimating the degree of business content loss consists of six nodes, the graph for relative business interruption loss of five nodes. The structures of the graphs are visualised in Fig. 3 and Fig. 4. The first parent node of each BN graph was set based on the strongest unconditional rank correlation between a predictor and the target variable (degree of content loss, relative interruption loss). This highest unconditional correlation coefficient exists for both constructed BNs for the variable-pair of water depth inside the building to the corresponding flood loss variable (Spearman's rank coefficient value, rho, for degree of content loss, rho: 0.37; for relative interruption loss, rho: 0.24). However, in the feature space for relative interruption losses, an equally strong correlation exists between the target and the indicator of emergency measures. This feature was identified by the CRF model as the most predictive for estimating relative interruption losses (Fig. 2.b) but was considered unimportant during the conditionalization of the BN, so the corresponding graph was constructed without it (Fig. 4). The variables for the remaining parent nodes were selected based on the strongest conditional ranking correlation using the CRF ranking as a guideline to prioritize the testing of potential parent nodes.

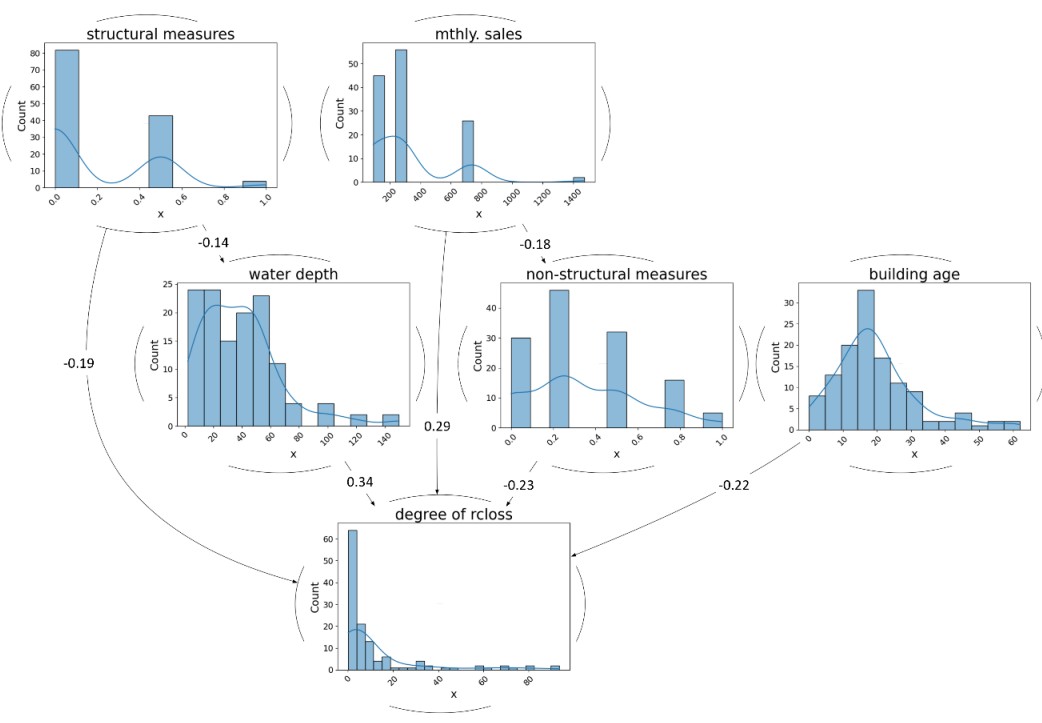

**Figure 3: Structure of the Bayesian Network for predicting the degree of business content loss (degree of rcloss). The values represent the rank correlation coefficients between the variables (rho).**

The predictors of flood losses and their assumed dependencies in the BN graphs are presented in the following:

- The degree of losses to business content and relative interruption losses, correlate with water depth in the shophouses (**water depth**). It is the predictor with the strongest rank correlation to both flood loss types (rho: 0.34 in Fig. 3, rho: 0.23

in Fig. 4) and was also previously identified as a relevant predictor by the CRF model. Rising water levels in the building directly increase the potential damage to low-lying goods, equipment and machinery (Kreibich et al., 2010; Chinh et al., 2015; Sieg et al., 2017). Apart from (non-)structural damages, the flooding of business premises itself or indirectly through power outages potentially leads to business interruptions (Kreibich et al., 2009; Sultana et al., 2018).

- High flow velocities (**flow velocity**) on the streets are associated with more severe business interruptions as indicated by

300 a correlation coefficient of 0.23 (Fig. 4) but are not important for modelling the degree of content losses. Business activities are potentially affected when high velocities hamper the transportation, such as by relocated objects blocking streets, or damage infrastructure, such as the energy systems (Bloch et al., 2012). Additionally, flow velocities have a direct effect on the water level in buildings by pressing water through openings in windows or doors, as also expressed in the BN graph for relative interruption losses of Fig. 4 (rho: 0.36). However, the missing impact of flow velocity on the degree of content

loss is explained by the high level of preparedness of HCMC's residents, such as the relocation of vehicles before potential

flooding (Chinh et al., 2016), whereas business activities, especially those of shops and small retailers, cannot or only partially be relocated to other premises.

- Age of the shophouse (**building age**) and degree of content loss have a negative relationship in the BN graph (rho: -0.22 in Fig. 3). The majority of shophouses in the HCMC samples were built in the last 30 years before the flood event, i.e. mainly between the 1980s and late 2000s. These "newer" shophouses reported the most severe content losses and can be explained by the strong urban pressure in these decades. The findings are confirmed by Downes and Storch (2014), Chinh et al. (2015) and Nguyen et al. (2016), who highlight that "newer" buildings in HCMC are more flood-exposed than "older" ones.

- The revenue from business operations (**mthly. sales**) is positively correlated with the degree of content loss in the respective BN graph as shown in Fig. 3 (rho: 0.29), but only a weak positive correlation exists to relative interruption losses. Monthly sales are seen as an indicator for the microbusiness size and its type of business content, as they reflect the heterogeneity among companies (Schoppa et al., 2020). The level of sales affect both exposure and vulnerability. Higher sales can increase exposure by driving expansion into risk-prone areas and requiring larger inventories, which are more susceptible to extreme weather events. The variable of monthly sales has a negative correlation with the uptake of non-structural precautionary measures in the graph for degree of content loss (rho: -0.18 in Fig. 3). This is theoretically explained by the connections within the data: businesses with limited revenues are more likely to acquire non-structural measures before the flood event, as loss of contents would have existential consequences for small retailers compared to more prosperous businesses.

- As shown by the BN graph in Fig. 3, the implementation of non-structural precautionary measures (**non-structural measures**) reduces the severity (degree) of content losses in microbusinesses (rho: -0.23 in Fig. 3). Though such measures are not relevant for modelling relative interruption losses. The impact of precautionary measures on reducing commercial content losses is well studied (Kreibich et al., 2007; Kreibich et al., 2010; Chinh et al., 2016; Sieg et al., 2017; Schoppa et al., 2020). Non-structural measures usually prevent water from infiltrating into the building, but not in all cases. For instance, Chinh et al. (2016) found that in Can Tho flood water can also come from the sewage system and thus bypass implemented precautionary measures. Consequently, there is no link with water depth in our model due to weak correlation between water depth and non-structural measures.

- The implementation of structural precautionary measures (**structural measures**) has mitigating effects on the severity of content and interruption losses in microbusinesses (rho: -0.19 in Fig. 3, rho: -0.11 in Fig. 4). The moderate dependencies in the BN graphs are in line with the findings of various studies, which highlight the usage of structural measures as an efficient individual precautionary measure (Scussolini et al., 2017; Trinh and Thanh, 2017; Du et al., 2020; Harish et al., 2023). The efficiency of these measures is represented in the BN graphs indirectly by lower water levels in the shophouses and directly in the flood loss variables, e.g., in elevated buildings, there is less chance that flood water will enter the building.

- A higher number of employees (**no. employees**) is linked with lower interruption losses in the respective BN graph (rho:
340       -0.13 in Fig. 4). Despite its rather weak negative rank correlation it improves the predictive accuracy of the BN model. The number of employees refers to the availability to human resources on which the business owner can draw on, which in turn affects the possibility to keep the business running during the flood event, for example, by relocating important business processes.

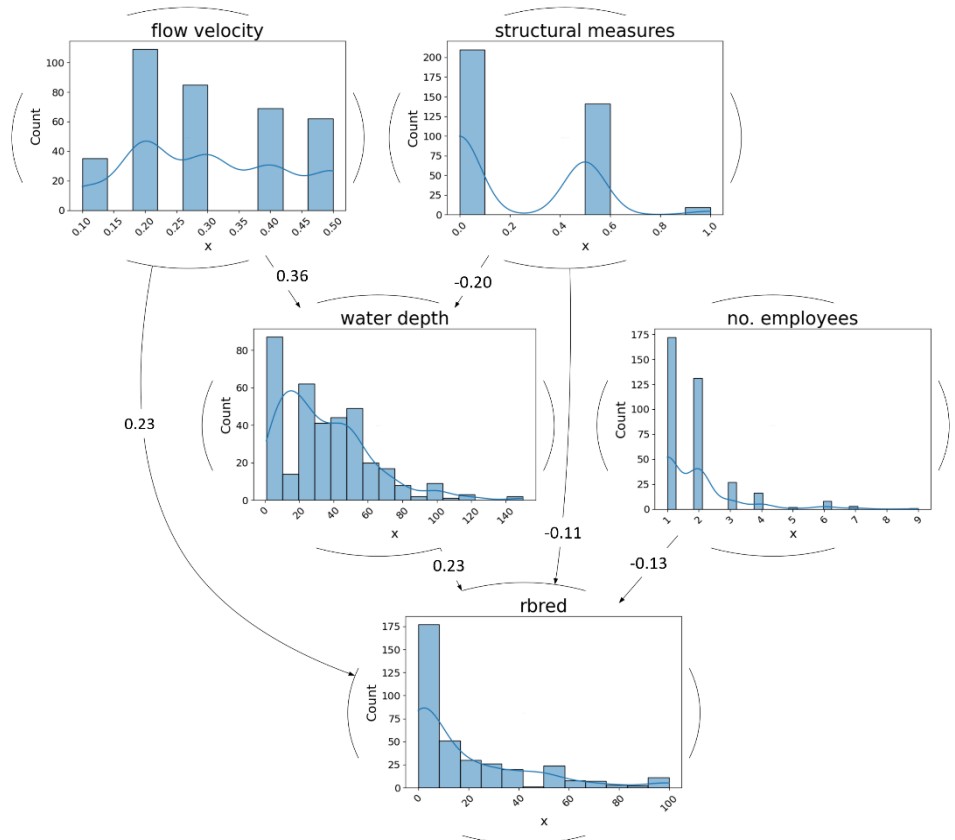

**Figure 4: Structure of the Bayesian Network for predicting relative business interruption losses (rbred). The values represent the rank correlation coefficients between the variables (rho).**

## 4.3 Flood loss model validation

### 4.3.1 Relative content loss

At the first glance, the modelling approach consisting of logistic regression and Bayesian Network seems to perform quite well when predicting relative content losses (MAE: 3.8 %, RMSE: 12.3 %). It marginally underestimates losses (MBE: -2.4 %) and has a remarkable low SMAPE of 16.3 % indicating a good level of precision. The mean value of the modelled relative content

losses is of similar magnitude as the observed loss ratios (observed mean: 4.7 %, predicted mean: 4.6 %), as shown also by the clustering of the data points in the lower value range in Fig. 5.a. However, the figure also illustrates that more severe losses to business content are consistently underestimated by the models.

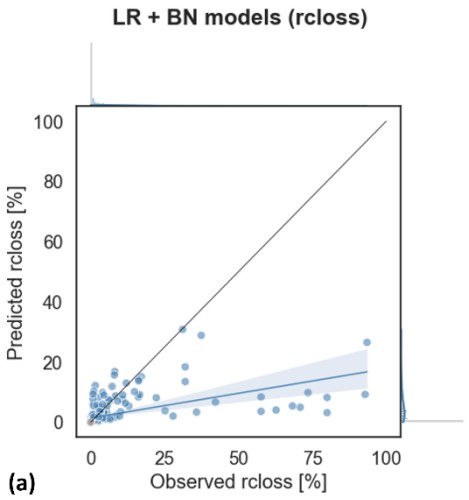
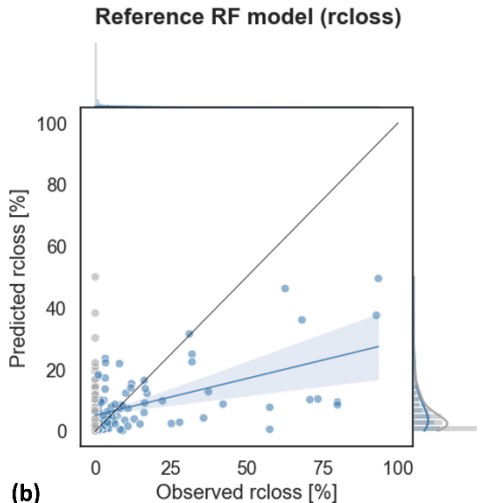

**Figure 5: Scatterplots of observed and modelled relative content losses (rcloss) to HCMC's microbusinesses for (a) the combination of logistic regression (LR) and Bayesian Network (BN) and (b) the reference Random Forest (RF) model used for benchmarking. The grey points represent the observations of zero-loss. The ML-based classifiers assigned to most cases an absence of content loss (zero-loss), thus only one grey point seems to be visualised in (a).**

The general well predictive performance of the modelling approach is caused by the usually low probability values for the chance of content loss. Having a critical look to the predicted probabilities of chance of content loss, it becomes clear that the observed small prediction bias is caused by the circumstance that the logistic regression estimated instances of chance of content loss usually as zero-loss cases. Thus, it assigns low probability of losses to most predictor combinations (see, the high share of cases predicted as zero-losses in the left half of Fig. 6.a).

The large number of observations of content loss wrongly predicted as zero-losses further illustrates this (see, False Positives in the lower left corner of Fig. 6.b); only 25 % of the experienced content losses (loss cases) are correctly predicted by the ML-classifiers (see, True Negatives in the lower right corner of Fig. 6.b).

As a consequence, most estimates for relative loss to business content are reduced by more than half as soon as they are multiplied with the predicted probabilities for chance of loss. In particular, the estimates of severe cases of content loss are reduced in their magnitudes. Furthermore, the ML-based classifiers could hardly distinguish between cases with an absence of loss (zero-loss) and small loss fractions (near zero-loss), which further deteriorated their calibration and performance.

In comparison to the modelling approach, the reference Random Forest model (Chinh et al., 2017) does not capture reported cases of zero-loss as such. This is shown when comparing the predicted values of zero-loss cases from the modelling approach (see, grey dots in Fig. 5.a) with the ones from the reference RF model (see, grey dots in Fig. 5.b). However, the general

predictive performance is only marginally worse (Table 2). The cross-validated RF-estimators have on average similar magnitudes in the RMSE (12.4 %) and MBE (1.3 %) as the modelling approach, but higher MAE (7.2 %) and SMAPE (78.9 %).

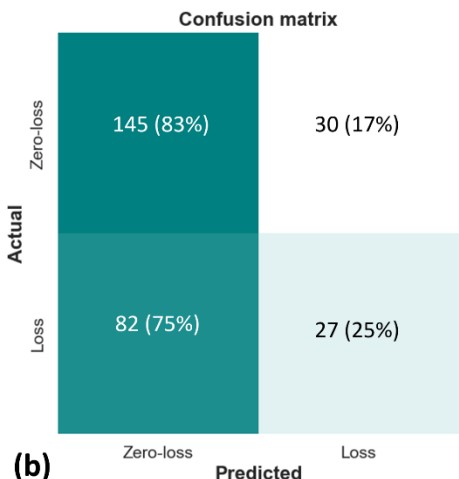

**Figure 6: (a) Distribution of observed (either 0.0 or 1.0) and predicted probabilities for chance of content loss. A vertical dashed line separates the observed and predicted cases of zero-loss from the observed and predicted loss cases. (b) The corresponding confusion matrix for chance of content loss. The values in front of the brackets are the sample numbers; values in the brackets the sample numbers normalized over the observations.**

### 4.3.2 Relative interruption loss

The cross-validation of the BN model for relative interruption losses results in an averaged MAE of 18.7 %, RMSE of 24.5 %, MBE of 0.17 % and SMAPE of 61.9 %. The modelled mean value in the interruption losses is almost equal to the observed mean of around 18.5 %, yet the variation in the observations is not well represented in the model estimates, as visualised in Fig. 7.a. Nearly all reported cases of interruption loss are predicted by the BN with loss fractions between 10 % and 40 %. This is much narrower compared to the variation seen in the reported loss ratios ranging between 0 % to 100 % decrease in business revenues. Additionally, the figure shows that more severe cases of interruption loss are underestimated by the BN, despite their rather frequent occurrence.

The reference RF model results in similar high prediction errors as the BN (Table 2). They particularly overestimate cases of zero- and near zero-loss and underestimate severe loss cases (Fig. 7.a and Fig. 7.b).

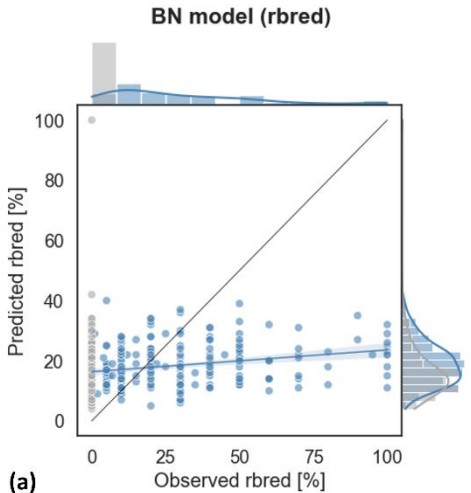

**Figure 7: Scatterplots for observed and modelled relative interruption losses (rbred) to HCMC's microbusinesses for (a) the Bayesian Network and (b) the reference Random Forest model used for benchmarking. The grey points represent the observations of zero-loss, i.e. the absence of interruption loss.**

## 4.4 Transferability of the flood loss models

In order to test the transfer capability, the interruption loss model calibrated on microbusinesses in HCMC was applied to predict interruption losses in Can Tho using a comparable survey dataset. The same procedure was applied to the models for content loss. However, the transferred logistic regression model was not able to capture the variation for chance of loss in the Can Tho samples. Thus, the study only presents the results of the transferred Bayesian Network model for interruption losses and the corresponding reference Random Forest model.

The generalisation ability of the BN model on the Can Tho samples results in similar prediction errors than during training on the HCMC samples, except for the SMAPE score. The transfer of the BN leads to a MAE of 17.9 %, RMSE of 23.5 %, MBE of 0.2 % and SMAPE of 23.2 %. The error scores show that the model's capacity to estimate interruption losses remains unchanged when transferred to Can Tho, in contrast to the transferability of the reference Random Forest model which resulted in a degraded performance (Table 2). These findings are reflected by cumulative distribution functions (CDFs) in Fig. 8.a and Fig. 8.b. The cumulative distributions shown in this study represent the change in the predictive accuracy of a model due to regional transfer. In other words, the CDFs provide insight into the extent to which a transferred flood loss model suffers from the different information contained in the Can Tho samples. The CDFs are shown in their normalized version to facilitate comparability despite different sample sizes.

The probability of the BN to predict a Can Tho sample precisely (prediction bias $< \pm 10$ %) remains unchanged (Fig. 8.a) but drops for the reference RF model from around 45 % (HCMC samples) to 25 % (Can Tho samples) (Fig. 8.b). The reference RF model underestimates interruption losses in nearly 90 % of the Can Tho samples, but in only 30 % of the HCMC samples

(Fig. 8.b). These findings show that the reference RF model is less transferable than the BN, despite both models perform similarly well in their calibration site (i.e., HCMC).

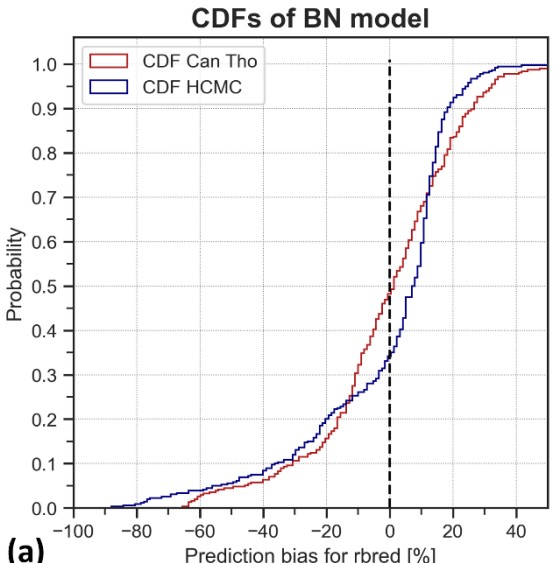 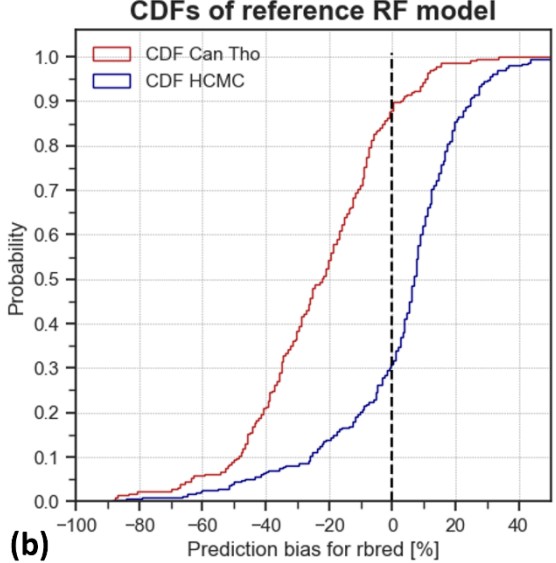

**Figure 8: Cumulative distribution function (CDF, normalized) of prediction errors for modelling business interruption losses (rbred)**
**in HCMC and in the transfer region, Can Tho. (a) CDF of the Bayesian Network performances; (b) CDF of the reference Random Forest model performance. The CDF for the reference RF model is cut by 50 % as no larger prediction errors exist.**

Transfer experiments on (Bayesian Network) flood loss models have highlighted that model transfer usually leads to a stagnation or drop in the model's performance, in particular, when the new conditions differ remarkably from the calibration
region (Schröter et al., 2014; Wagenaar et al., 2018). However, there was no drop in the performance of the BN model performance when transferred across regions. This is due to very similar local conditions between the calibration (HCMC) and transfer site (Can Tho). These local conditions are reflected in the similar predictor ranges and distributions of both survey datasets (see, Supplementary Information Fig. S2). Additionally, the high heterogeneity in the HCMC samples, in particular, in the hydrological, building- and business-related predictors, has the potential to increase the model robustness for new study
sites (Wagenaar et al., 2018).

**Table 2: Model validation of flood loss models in HCMC and in the transfer region (Can Tho). The different sample sizes are due to the differences in the number of cases reported and the way in which incomplete samples are treated in the models. rcloss: relative loss to business content, rbred: relative loss due to business interruption, LR: probabilistic logistic regression, BN: Bayesian Network, RF: reference Random Forest, x: not applicable**

| | Model validation [sample size] | MAE [%] | RMSE [%] | MBE [%] | SMAPE [%] |
|---|---|---|---|---|---|
| **HCMC** | | | | | |
| *rcloss* | LR + BN [284] | **3.8** | 12.3 | -2.4 | 16.3 |
| | RF [284] | 7.2 | 12.4 | 1.3 | 78.9 |
| *rbred* | BN [361] | 18.7 | 24.5 | 0.17 | 61.9 |
| | RF [314] | **16.4** | 21.8 | 1.7 | 58.6 |
| **Can Tho (transfer region)** | | | | | |
| *rcloss* | LR + BN [266] | x | x | x | x |
| | RF [266] | **13.5** | 19.6 | 0.8 | 75.0 |
| *rbred* | BN [313] | **17.9** | 23.5 | 0.2 | 23.2 |
| | RF [267] | 25.7 | 32.6 | -23.5 | 41.1 |

## 4.5 Applicability, limitations and uncertainties

Reliable flood loss models are essential tools for stakeholders and practitioners across multiple sectors, including insurance, urban planning, flood risk management, and climate adaptation decision-making. The flood loss models presented in this study specifically address the economic impacts of flooding on microbusinesses in Vietnamese cities, focusing on business interruptions and content losses. To our knowledge, this type of economic sector is underrepresented in risk management and the proposed models can advance decision making with a focus on this sector. By representing key drivers of loss as graph structures, the models offer an interpretable and transparent framework for understanding how various factors contribute to flood-related damages. The models are based on non-parametric Bayesian Networks, which enable probabilistic estimation of flood losses while explicitly quantifying uncertainty in both data and model formulation. This feature makes the models particularly robust, allowing for transparent assessment of risk and greater confidence in the results. Unlike traditional deterministic models, the Bayesian approach ensures flexibility in handling incomplete or uncertain data, which is a common challenge in flood loss estimation. The combination of an interpretable model structure and transparent uncertainty quantification opens the door for operationalizing this modelling approach in practical settings. It provides stakeholders with a clearer understanding of how flood losses are calculated, promoting trust and facilitating decision-making. Furthermore, the model's ability to function effectively even with missing or limited data enhances its transferability to similar geographic regions and contexts. This adaptability is particularly valuable for expanding its application in data-scarce environments or in rapidly urbanizing areas where flood risks are evolving.

Despite these advantages, the models rely on empirical post-event survey datasets and have certain limitations. For instance, the sample was obtained voluntarily, which may introduce selection bias. The study focused on frequently flooded regions, including both well-established city areas and newly urbanized zones, to represent the city's expansion. However, the absence of official loss data prevents validation of the reported figures, particularly given the potential for under-reporting. In addition to the biases in the survey data, the modelling results indicate high uncertainty in reconstructing flood losses from survey data. One possible further analysis would be comparing the model estimates with other studies. However, comparability is limited by the fact that in contrast to our object-level modelling, state-of-the-art flood loss modelling in low- and middle-income countries is mainly carried out on meso- or macroscale (Booij, 2004; Aerts et al., 2020; Tierolf et al., 2021), with commercial losses reported only in absolute values (Wijayanti et al., 2017; Patankar, 2019; Tsinda et al., 2019) and often without validation (Ke et al., 2012; Patankar and Patwardhan, 2015; Yang et al., 2016).

We did not observe an increase in model uncertainty for the Bayesian Network model for interruption losses due to the regional transfer. Furthermore, the mean values of the empirical interruption losses are for both regions (HCMC, Can Tho) within the uncertainty ranges (within 95 % confidence interval. However, as seen above, the majority of interruption-related losses are underestimated by the flood loss models remarkably.

The example of regional transfer illustrates the potential of non-parametric, continuous Bayesian Network models compared to Random Forest models. However, since the transfer capability was validated for only one case study, there is a need to calibrate and validate the loss models under further local and temporal conditions.

## 5 Conclusions

We proposed a first approach to estimate flood losses to microbusinesses by combining expert knowledge with survey data of flood-affected microbusinesses from HCMC and Can Tho in Vietnam. A Conditional Random Forest model was applied to obtain the main drivers of content and interruption losses from a set of heterogeneous samples and potential predictors which are partly correlated to each other. The identified drivers were used to calibrate knowledge-based probabilistic loss models consisting of non-parametric, continuous Bayesian Networks and logistic regression. The findings of this study indicate that information on business revenues from monthly sales and production, building age, and hydrological characteristics of the flood is crucial in estimating content and interruption losses for microbusinesses.

The probabilistic flood loss models were calibrated and validated against reported flood losses in HCMC and in a transfer case study of Can Tho. The study resulted in interpretable and transferrable probabilistic flood loss models for predicting content losses and business interruption losses to microbusinesses. In addition, the models are openly provided and integrating them to flood risk assessments has the potential to advance risk management decision making with a focus on microbusinesses.

*Data and Code Availability*. The survey data will be made openly available in the HOWAS21 database (https://howas21.gfz-potsdam.de/) after an embargo of three years after the end of the project (in 2027). The data can be accessed in the meantime from the authors. Source code (python) is openly available at https://github.com/A-Buch/flood-loss-models-4-HCMC/tree/microbusiness-paper. The Bayesian Network flood loss models are created with the PyBanshee toolbox (Koot et al., 2023, https://github.com/mike-mendoza/py_banshee.git); the Conditional Random Forest models are based on the R package partykit (Hothorn and Zeileis, 2015, https://www.jmlr.org/papers/v16/hothorn15a.html).

*Author contributions.* Conceptualisation: AB, DP, NS; Data curation, formal analysis and visualization: AB; Methodology: AB, DP, KRS, HK, NS; Supervision: DP, NS; Writing – original draft: AB; Writing – review & editing: AB, DP, KRS, HK, NS

*Competing interests.* Co-author Heidi Kreibich is member of the editorial board of Natural Hazards and Earth System Sciences.

*Funding.* The data collection was undertaken by project DECIDER (grant nos. 01LZ1703G and 01LZ1703A) and WISDOM II ("Water related Information System for a Sustainable development of the Mekong Delta") funded by the BMBF (German Ministry for Education and Research). Nivedita Sairam is funded by project HI-CliF (grant no. 01LN2209A) funded by the BMBF.

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
