# Peer review of "Modelling Flood Losses to Microbusinesses in Ho Chi Minh City, Vietnam"

_EGUsphere, 2024_

## Author Comment (AC1)

**Response to Referee 1:**

Below, we respond (**R**) to the more general and major comments (**C**) of the reviewer. The changes in the revised text are mentioned in the response letter in *italics* along with line numbers referring to the revised manuscript (which is not uploaded yet). The revised manuscript includes one more figure and an additional subsection, hence the figure/section numbers of the preprint [*preprint: figure/section number*] are included in the response letter, when they differ between both manuscript versions.

**General comments**

**C1.0:** This paper aims to identify the drivers of flood losses in microbusinesses by employing a Conditional Random Forest on survey data collected from microbusinesses in Ho Chi Minh City in Vietnam. Based on the drivers identified, probabilistic loss models (non-parametric Bayesian Networks) were developed using a combination of data-driven and expert based model formulation. The transferability of these models was assessed by applying data from a different city to evaluate their broader applicability. I have read the paper with great interest, and the main objective addressed by the manuscript is within the scope of the journal. Nevertheless, major revisions are necessary to make a few points clearer and I recommend accepting it only after these revisions.

**R1.0:** We thank the referee for taking the time and effort to provide comprehensive feedback on our manuscript. We have implemented almost all the suggestions, otherwise, an explanation is provided in this response letter. In particular, the data and methodology sections in the revised version of our manuscript are adapted and extended significantly.

**Major comments**

**C1.1:** I believe the paper could benefit from separating the results from the discussion to enhance clarity. As someone without extensive expertise in ML algorithms, I found it challenging at time to connect the information in the figures and tables with the text. For instance, on pages 10 and 11, Figure 3 is only referenced once, and while the authors discuss correlations among variables, they do not always provide specific numerical values from the figures. The paper contains a substantial amount of results, which makes it difficult to easily connect the text with the accompanying figures and tables. Please, check the figures axes names. Sometime you start with capital letter and sometimes with small

**R1.1:** In the revised manuscript, we made an effort to better link the figures and tables to the text. In particular,

1. To better highlight the important aspects in the text, namely the chosen factors and their dependencies in the BN graphs. The rank correlation coefficients from

Figure 3 and 4 *[preprint: Figure 2 and 3]* are now consistently mentioned in the text, for example (lines 287:288):

*"It is the predictor with the strongest rank correlation to both flood loss types (rho: 0.34 in Fig. 3, rho: 0.23 in Fig. 4) and was also previously identified as a relevant predictor by the CRF model."*

2. To ensure traceability of the construction process of the BN graphs, we substantiated the selection of each factor and its dependencies in the graphs with findings from selected literature, for example (lines 325:330):

*"The moderate dependencies in the BN graphs are in line with the findings of various studies, which highlight the usage of structural measures as an efficient individual precautionary measure (Scussolini et al., 2017; Trinh and Thanh, 2017; Du et al., 2020; Harish et al., 2023). The efficiency of these measures is represented in the BN graphs indirectly by lower water levels in the shophouses and directly by less severe flood losses, e.g. in elevated buildings, there is less chance that flood water will enter the building during a flood event."*

3. We want to set these results directly into context with findings from other flood loss studies and the regional conditions of the study area. Separating results and discussion section may lead to repetitions and increasing the manuscript length. Hence, Sect. 4.1, 4.3 and 4.4 present relevant study findings in the beginning and discusses them in the end of the respective subsection. In this way the results are separated from their discussion within each subsection. We hope this improves the clarity of the manuscript.

**C1.2:** Additionally, I recommend simplifying certain figures (e.g. Figure 5) or providing more detailed descriptions of them in the text. Given that this journal focuses on natural hazards research and is accessed by readers who may not be experts in ML algorithms, a clearer structure with separate results and discussion would aid.

**R1.2**: We fully agree with the referee in regard to Figure 6 *[preprint: Figure 5]*. Thus, we improved this figure as well as Figure 8 *[preprint: Figure 7]*. In addition, the revised version of the manuscript provides more description of them. Regarding Figure 8 we would like to refer to our response **R1.4**. Please find below the modified description of Figure 6 and its description in Sect. 4.3.1.

Adapted figure description of Figure 6 *[preprint: Figure 5]*:
*"(a) Distribution of observed (either 0.0 or 1.0) and predicted probabilities of chance of content loss from the ML-based classifiers. A vertical dashed line separates the observed and predicted cases of zero-loss from the observed and predicted loss cases. (b) The corresponding confusion matrix for chance of content loss. The values in front of the brackets are the sample numbers; values in the brackets the sample numbers normalized over the observations."*

Adapted sentence in Sect. 4.3.1 (lines 355:358):

*"The predicted probabilities for chance of content loss show that the observed small prediction bias is caused by the circumstance that the logistic regression estimated instances of chance of content loss usually as zero-loss cases. Thus it assigns low probability of losses to most predictor combinations (see, the high share of cases predicted as zero-losses in the left half of Fig. 6.a)."*

As mentioned in **R1.1**, we have made an effort to improve the clarity of the manuscript by restructuring the sections separating the discussion from the results and not introducing a separate discussion section.

**C1.3**: In the methodology section, I suggest that the authors provide further clarification in certain areas. For example, in the abstract, content losses are reported 317 and business interruption losses as 361. However, in the section presenting the data is noted that 250 responses were collected resulting in 397 loss records in the HCMC and for Can Tho, responses were received from 373 microbusiness, of which 313 provided information on losses. It is unclear how these numbers were derived and calculated. So, please check the numbers and a more detailed explanation of the methods used would be helpful.

**R1.3:** In fact, we did not adequately explain the different sample sizes in the manuscript and have therefore improved Sect. 2.2.1 and 2.2.2 *[preprint: Sect. 2.1 and 2.2]* in the revised version of the manuscript, as well as Sect. S1 in the Supplementary Information. In the following, we provide further information about the pre-processing steps and the returned sample sizes for the two survey datasets.

HCMC dataset (Sect. 2.2.1, *[preprint: Sect. 2.1]*):
Some of the 250 interviewees in Ho Chi Minh City did not provide information on both flood loss events. In addition, information about the flood loss type (interruption loss or content loss) was missing or could not be derived resulting in 361 samples for interruption losses and 317 for content losses. The section now clearly explains how the different sample sizes for relative interruption losses and for relative content losses were derived from the HCMC dataset (lines 126:134):

*"However, not every interviewee provided information for both events, which leads to a number of 397 loss records in the HCMC dataset. Each record in this dataset comprises information about one or two types of flood losses experienced during an event. In detail, 361 samples of the loss records contain information about business interruption losses reported as relative values (e.g. reduced sales and production), while a similar sample size comprises flood losses to business content but reported as monetary values (e.g. to furniture, electrical devices, stored products and vehicles). Conversion to relative scales reduced the number of content loss samples by using exposure information about the value of business content, as described in the Supplementary Information, Sect. S1. Consequently, the sample size referring to relative content losses (n=317) is smaller than for relative interruption losses (n=361).*

*Hereinafter both types of relative flood losses are referred to as flood loss variables (Table 1)."*

The above citation mentions that the building values were needed to convert the reported content losses to relative scales, however, in some records information about the building value was missing. Therefore, the final sample size for relative content losses is reduced. Two sentences explaining this aspect are added to the revised version of the Supplementary Information (Sect. S1, lines 21:24):

*"However, some interviewees did not report the building value and thus relative content losses were not calculated for these records, resulting in a reduced number of 317 records for relative contents losses. Since business interruption losses were already queried as relative values in the HCMC survey, their number of 361 records remained unchanged."*

Can Tho dataset (Sect. 2.2.2 *[preprint: Sect. 2.2]*):
The revised version of the manuscript now points out that relative content losses were calculated differently for the Can Tho survey compared to the HCMC survey (lines 145:150):

*"The survey was undertaken in January-February 2012 and received responses from 373 microbusinesses out of which 313 furnished information on losses to business content and due to business interruption. The questionnaire is comparable to the survey undertaken in HCMC, with the exceptions that each interviewee reported only about the most severe flood event during 2011 and provided information about the value of the business content. The latter information was used to calculate relative content losses. Furthermore, the microbusinesses' locations were not queried. All other pre-processing steps were the same as for the HCMC data."*

**C1.4**: Additionally, presenting the equations for the error formulas mentioned in lines 147–150 would enhance clarity. There are also methods referenced in the results that are not described in the methodology section. For instance, cumulative distribution functions (CDFs) are discussed in line 357, yet they are not explained in the methods. Including these details would improve transparency and ensure a more complete understanding of the approach used.

**R1.4:** We included the equation for the error formulas in a revised version of the Supplementary Information (see, Table S1). A brief explanation of cumulative distribution functions is added at the end of Sect. 3.2.2 in the revised manuscript (lines 231:237):

*"Cumulative distribution functions (CDFs) were used to visualise the results of the transfer experiment. The cumulative distributions shown in this study (Fig. 8.a and 8.b) represent the change in the predictive accuracy of a model due to regional transfer. In other words, the CDFs provide insight into the extent to which a transferred flood loss model suffers from the different information contained in the Can Tho samples. For example, Fig. 8.b shows that the reference RF underestimates interruption losses in nearly 90 % of the Can Tho samples, but in only 30 % of the HCMC samples. The*

*CDFs are examined in their normalised versions to keep the plots of the cumulative distributions comparable, regardless of their different sample sizes."*

**C1.5:** In the introduction, the authors place emphasis on the case study to motivate the analysis. It may be beneficial to move the detailed description of the case study to a separate section, allowing the introduction to focus more directly on the research gaps. This would help to clearly establish the broader motivation and context for the study before delving into the specifics of the case study.

**R1.5**: Thank you for this suggestion. In the revised version of the manuscript, we moved the description of the case study to a new subsection of Sect. 2. Furthermore, we renamed Sect. 2 from "*Data – Post-flood survey of microbusinesses*" to "*Domain and data*" (line 76) with subsections covering the case study and the post-flood survey datasets.  We hope this will make the broader motivation and context for the study clearer.

**C1.6:** The discussion and conclusion sections could be enhanced by further exploring how the findings may be utilized by other experts and their implications for flood risk management. Expanding on these aspects would clarify the broader relevance of the outputs.

**R1.6:** To illustrate the broader relevance of the study, we have improved the discussion and conclusions by highlighting the need of flood loss models to stakeholders and how reliable loss model predictions can improve decision making in terms of risk assessment.

---

## Author Comment (AC2)

Below, we respond (**R**) to the more general and specific comments (**C**) of the reviewer. The changes in the revised text are mentioned in the response letter in *italics* along with line numbers referring to the revised manuscript (which is not uploaded yet). The revised manuscript includes one more figure and an additional subsection, hence the figure/section numbers of the preprint [*preprint: figure/section number*] are included in the response letter, when they differ between both manuscript versions.

**General comments**

**C2.0:** The manuscript addresses the topic of flood losses to microbusinesses in low- and middle-income countries and introduces an application of Conditional Random Forests for feature selection and Bayesian Networks to model flood losses, bridging gaps in existing methods that primarily cater to larger firms or macro-level analyses. In addition, it provides empirical evidence for key drivers of flood losses (e.g., water depth, building age, monthly revenue) and evaluates the models' transferability to another city (Can Tho), highlighting its potential for broader regional application. The approach is good in general; however, the manuscript should provide more details for readers to understand and reproduce this approach in other case studies.

**R2.0:** We thank the referee for taking the time and effort to provide comprehensive feedback on our manuscript. In the revised version of the manuscript, we have implemented almost all the suggestions, otherwise, an explanation is provided in this response letter. Due to the length of the manuscript, we did not explain the individual pre-processing steps applied on the survey datasets in detail, but rather provided a brief overview, as mentioned in **R1.3** in the first response letter. We had also removed parts of the methodology section (e.g. the flowchart) in the manuscript to keep it as short as possible, however, we included these parts again in the revised versions of the manuscript and the Supplementary Information (e.g. flowchart in Fig. S1).

**Specific comments**

**C2.1:** Provide more explanation about machine learning techniques of "Bayesian Network" and "Conditional Random Forest" and briefly explain why these methods suit the study.

**R2.1:** Indeed we have not explained the models in-depth due to the already rather large extent of the manuscript. As suggested, we have added brief explanations for both models and pointed out the suitability of the chosen methods compared to other ML-based approaches.

In the revised version of the manuscript, we modified and extended following explanations of Conditional Random Forests in Sect. 3.1.1 (lines 180:187):

*"The set of candidate predictors presented in Table 1 exhibits a moderate to high degree of multicollinearity, for instance, the flood-related features are strongly correlated to each other. For this reason, Conditional Inference Trees were applied to account for these correlations during feature selection. Conditional Inference Trees were initially introduced by Hothorn et al. (2006) and extended by Strobl et al. (2007) to an ensemble of trees, a so-called Conditional Inference Random Forest (CRF). Each tree is grown only by a subset of features, which were identified before as significant based on their p-values (Hothorn et al., 2006). By this approach predictive features are identified, despite their potential collinearity to other candidate predictors. The choice of an unbiased version of the permutation-based feature importance method – namely Conditional Permutation Importance - further reduces the chance of biased importance scores for correlated features (CPI, Debeer and Strobl, 2020)."*

In the revised version of the manuscript, we added following explanations to Bayesian Networks (BNs) in Sect. 3.2.2 (lines 210:216):

*"They are better regional transferable than other ML-based models, such as regularized linear regressions, since BNs can be updated with new data and applied on incomplete information. Furthermore, they have the benefit of explicitly representing the dependency structures, quantifying uncertainty and the possibility of including expert knowledge alongside data. In more detail, the dependency structure of a BN represents (assumed) causal relations between variables, these dependencies can be set based on knowledge or logical conclusions. For example, the relation that less flood water infiltrates into a building if structural measures were taken beforehand is represented in the BN graph of Fig. 3 by a negative correlation (rho: -0.14) between both variables."*

However, the manuscript does not deal with the conceptual basis of Bayesian networks - the Bayes theorem - or other aspects of them. For a more in-depth understanding, we would like to refer for CRF models to the work of Strobl et al. 2007 and Levshina 2020; for non-parametric Bayesian Network flood loss models to the work of Paprotny et al. 2021. In particular, the benefits of BNs briefly mentioned in the manuscript (lines 211-212) - "*explicitly representing the dependency structures, quantifying uncertainty and the possibility of including expert knowledge alongside data*" - are described in more detail in the latter study.

CRF:

Levshina, N.: Conditional Inference Trees and Random Forests, in: A Practical Handbook of Corpus Linguistics, edited by: Paquot, M. and Gries, S. Th., Cham: Springer, 611–643, https://doi.org/10.1007/978-3-030-46216-1_25, 2020.

Strobl, C., Boulesteix, A.-L., Zeileis, A., and Hothorn, T.: Bias in random forest variable importance measures: Illustrations, sources and a solution, BMC Bioinformatics, 8, 25, https://doi.org/10.1186/1471-2105-8-25, 2007.

BN:

Paprotny, D., Kreibich, H., Morales-Nápoles, O., Wagenaar, D., Castellarin, A., Carisi, F., Bertin, X., Merz, B., and Schröter, K.: A probabilistic approach to estimating residential losses from different flood types, Nat. Hazards, 105, 2569–2601, https://doi.org/10.1007/s11069-020-04413-x, 2021

**C2.2:** Case study selection: The manuscript justifies the choice of Ho Chi Minh City and Can Tho as case studies based on their high flood risk and economic importance. However, additional clarification is recommended (e.g., explain how these cities represent other flood-prone urban areas in Vietnam or Southeast Asia).

**R2.2:** This is a good point to further detail the choice of both cities in regard to their economic and topographic similarity to other flood-prone urban areas. The rapid economic developments in Southeast Asian states in the last decades are partly driven by the operations of micro-, small- and medium-sized businesses. The economic sector in urban areas of Vietnam, not only in HCMC and Can Tho, consists to a large share of those smaller businesses. The following sentence illustrates this aspect (Sect. 2.2.1 *[preprint: Sect. 2.1]*, lines 124-125) "The presented shares of the business sectors in the HCMC survey are representative for entire Vietnam (General Statistics Office 2018)" - HCMC is economically representative to other urban areas in Vietnam. Moreover, most of Vietnam's economic centres are located on flat terrain with access to the coast or a major river. Thus, they are similar flood-prone as HCMC and Can Tho, however, their type of flooding might differ (fluvial, pluvial, coastal). In the revised version of the manuscript, we adapted the following sentences to make the topographic similarity of HCMC to other urban areas more explicit (Sect. 2.1 *[preprint: Sect. 1]*, lines 79-81):

"*Similar to other SE-Asian metropolises, HCMC lies in a river delta area close to the coast. These densely populated, flat, riverine and coastal regions experience regular flooding in particular during the rainy season (Garschagen, 2015; Tierolf et al., 2021; Nguyen et al., 2021).„*

**C2.3:** Defining sample size: The manuscript mentions the number of surveyed microbusinesses but lacks a detailed rationale for the sample size determination (What statistical considerations or sampling techniques were used to determine the sample size?).

**R2.3:** In fact, we did not adequately explain the different sample sizes in the manuscript and have therefore added the following sentences to Sect. 2.2.1 *[preprint: Sect. 2.1]* (lines 126:129):

"*In order to achieve a reasonable representation of HCMC, we selected the districts with the most frequent flood risk and also heterogeneity in socio-economic conditions. Within each district, the shophouses were chosen randomly. The sample size in each district was not chosen based on statistical considerations, but on recommendation from local experts.*"

**C2.4:** Ethical application for surveys: Ethical considerations for conducting surveys need to be explicitly addressed to ensure transparency and compliance with research standards.

**R2.4:** Thank you for the very crucial comment. The survey resulted in anonymous data from flood affected businesses. To connect the data of the interviews with flood characteristics, the location of the interviewed businesses was recorded. We decided to show in a GIS map (**R2.6**) only the rough geolocations of the microbusinesses to protect the anonymity of the interviewed businesses. The data stored and handled is conform to data privacy and data protection regulations.

**C2.5:** A flowchart summarizing the methodological workflow (from data collection to modeling and validation) can improve clarity for readers unfamiliar with machine learning or Bayesian methods.

**R2.5:** We moved the flowchart (Fig. S1) to the revised version of the Supplementary Information to not further extend the manuscript length. We linked the main steps, visualized by the flowchart, to the text in the data and methodology section.

**C2.6:** A GIS map that shows locations of surveyed microbusinesses can provide clear context and can be combined with flood hazard maps to give an overview of hazard and impact, which can be suitable for visualization and dissemination.

**R2.6**: We added a GIS map in the revised manuscript (Fig. 1) showing the rough locations of the microbusinesses surveyed in HCMC. The entire urban area of HCMC as well as suburban districts are located on low-lying terrain, thus, in combination with many open channels, the majority of HCMC's districts are facing a high flood risk. We tried to illustrate this aspect in the map by adding information about the low-lying areas alongside the rough locations of the microbusinesses. Besides the figure, we added the following text to the revised manuscript to explain this aspect (lines 134:137):

*"Figure 1 visualizes the rough locations of the microbusinesses surveyed in HCMC. However, their exact geolocations are not shown to protect the anonymity of the interviewees. Furthermore, the map shows that all surveyed microbusinesses are located on low-lying terrain often in short distance to a larger channel or tributary river."*

---

## Author Response (AR1)

Below, we respond (**R**) to the more general and major comments (**C**) of the reviewer. The changes in the revised text are mentioned in the response letter in *italics* along with line numbers. Furthermore, the revised manuscript includes one more figure (Fig. 1) and an additional subsection (Sect. 2.1).

General comments

**C1.0:** This paper aims to identify the drivers of flood losses in microbusinesses by employing a Conditional Random Forest on survey data collected from microbusinesses in Ho Chi Minh City in Vietnam. Based on the drivers identified, probabilistic loss models (non-parametric Bayesian Networks) were developed using a combination of data-driven and expert based model formulation. The transferability of these models was assessed by applying data from a different city to evaluate their broader applicability. I have read the paper with great interest, and the main objective addressed by the manuscript is within the scope of the journal. Nevertheless, major revisions are necessary to make a few points clearer and I recommend accepting it only after these revisions.

**R1.0:** We thank the referee for taking the time and effort to provide comprehensive feedback on our manuscript. We have implemented almost all the suggestions, otherwise, an explanation is provided in this response letter. In particular, the data and methodology sections in the revised manuscript are adapted and extended significantly.

Major comments

**C1.1:** I believe the paper could benefit from separating the results from the discussion to enhance clarity. As someone without extensive expertise in ML algorithms, I found it challenging at time to connect the information in the figures and tables with the text. For instance, on pages 10 and 11, Figure 3 is only referenced once, and while the authors discuss correlations among variables, they do not always provide specific numerical values from the figures. The paper contains a substantial amount of results, which makes it difficult to easily connect the text with the accompanying figures and tables. Please, check the figures axes names. Sometime you start with capital letter and sometimes with small

**R1.1:** In the revised manuscript, we made an effort to better link the figures and tables to the text. In particular,

1. To better highlight the important aspects in the text, namely the chosen factors and their dependencies in the BN graphs. The rank correlation coefficients from Figure 3 and 4 are now consistently mentioned in the text, for example (lines 290:291):
   *"It is the predictor with the strongest rank correlation to both flood loss types (rho: 0.34 in Fig. 3, rho: 0.23 in Fig. 4) and was also previously identified as a relevant predictor by the CRF model."*

2. To ensure traceability of the construction process of the BN graphs, we substantiated the selection of each factor and its dependencies in the graphs with findings from selected literature, for example (lines 328:333):

*"The moderate dependencies in the BN graphs are in line with the findings of various studies, which highlight the usage of structural measures as an efficient individual precautionary measure (Scussolini et al., 2017; Trinh and Thanh, 2017; Du et al., 2020; Harish et al., 2023). The efficiency of these measures is represented in the BN graphs indirectly by lower water levels in the shophouses and directly in the flood loss variables, e.g. in elevated buildings, there is less chance that flood water will enter the building."*

3. We want to set these results directly into context with findings from other flood loss studies and the regional conditions of the study area. Separating results and discussion section may lead to repetitions and increasing the manuscript length. Hence, Sect. 4.1, 4.3 and 4.4 present relevant study findings in the beginning and discusses them in the end of the respective subsection. In this way the results are separated from their discussion within each subsection. We hope this improves the clarity of the manuscript.

**C1.2:** Additionally, I recommend simplifying certain figures (e.g. Figure 5) or providing more detailed descriptions of them in the text. Given that this journal focuses on natural hazards research and is accessed by readers who may not be experts in ML algorithms, a clearer structure with separate results and discussion would aid.

**R1.2**: We fully agree with the referee in regard to Figure 6. Thus, we improved this figure as well as Figure 8. In addition, the revised manuscript provides a more detailed description of them. Regarding Figure 8 we would like to refer to our response **R1.4**. Please find below the modified description of Figure 6 and its description in Sect. 4.3.1.

Adapted figure description of Figure 6:
*"(a) Distribution of observed (either 0.0 or 1.0) and predicted probabilities for chance of content loss. A vertical dashed line separates the observed and predicted cases of zero-loss from the observed and predicted loss cases. (b) The corresponding confusion matrix for chance of content loss. The values in front of the brackets are the sample numbers; values in the brackets the sample numbers normalized over the observations."*

Adapted sentence in Sect. 4.3.1 (lines 358:361):
*"Having a critical look to the predicted probabilities of chance of content loss, it becomes clear that the observed small prediction bias is caused by the circumstance that the logistic regression estimated instances of chance of content loss usually as zero-loss cases. Thus, it assigns low probability of losses to most predictor combinations (see, the high share of cases predicted as zero-losses in the left half of Fig. 6.a)."*

As mentioned in **R1.1**, we have made an effort to improve the clarity of the manuscript by separating the discussion from the results and not introducing a separate discussion section.

**C1.3**: In the methodology section, I suggest that the authors provide further clarification in certain areas. For example, in the abstract, content losses are reported 317 and business interruption losses as 361. However, in the section presenting the data is noted that 250 responses were collected resulting in 397 loss records in the HCMC and for Can Tho, responses were received from 373 microbusiness, of which 313 provided information on

losses. It is unclear how these numbers were derived and calculated. So, please check the numbers and a more detailed explanation of the methods used would be helpful.

**R1.3:** In fact, we did not adequately explain the different sample sizes in the manuscript and have therefore improved Sect. 2.2.1 and 2.2.2 in the revised version of the manuscript, as well as Sect. S1 in the Supplementary Information. In the following, we provide further information about the pre-processing steps and the returned sample sizes for the two survey datasets.

HCMC dataset (Sect. 2.2.1):
Some of the 250 interviewees in Ho Chi Minh City did not provide information on both flood loss events. In addition, information about the flood loss type (interruption loss or content loss) was missing or could not be derived resulting in 361 samples for interruption losses and 317 for content losses. The section now clearly explains how the different sample sizes for relative interruption losses and for relative content losses were derived from the HCMC dataset (lines 133:142):

*"However, not every interviewee provided information for both events, which leads to a number of 397 loss records in the HCMC dataset. Each record in this dataset comprises information about one or two types of flood losses experienced during an event. In detail, 361 samples of the loss records contain information about business interruption losses reported as relative values (e.g. reduced sales and production), while a similar sample size comprises flood losses to business content but reported as monetary values (e.g. to furniture, electrical devices, stored products and vehicles). The conversion of the latter loss type to relative scales reduced it to 317 samples (relative content loss) by using the value of business content as exposure information (see, Supplementary Information Sect. S1). Consequently, the sample size referring to relative content losses (n=317) is smaller than for relative interruption losses (n=361). Hereinafter both types of relative flood losses are referred to as flood loss variables (Table 1)."*

The above citation mentions that the building values were needed to convert the reported content losses to relative scales, however, in some records information about the building value was missing. Therefore, the final sample size for relative content losses is reduced. Two sentences explaining this aspect are added to the revised version of the Supplementary Information (Sect. S1, lines 21:24):

*"However, some interviewees did not report the building value and thus relative content losses were not calculated for these records, resulting in a reduced number of 317 records for relative contents losses. Since business interruption losses were already queried as relative values in the HCMC survey, their number of 361 records remained unchanged."*

Can Tho dataset (Sect. 2.2.2):
The revised version of the manuscript now points out that relative content losses were calculated differently for the Can Tho survey compared to the HCMC survey (lines 154:159):

*"The survey was undertaken in January-February 2012 and received responses from 373 microbusinesses out of which 313 furnished information on losses to business content and due to business interruption. The questionnaire is comparable to the survey undertaken in HCMC, except that each interviewee reported only about the most severe flood event during 2011 and provided information about the value of their business content. The latter was used to calculate relative content losses. Furthermore, the microbusinesses' locations were not queried. All other pre-processing steps were the same as for the HCMC data."*

**C1.4**: Additionally, presenting the equations for the error formulas mentioned in lines 147–150 would enhance clarity. There are also methods referenced in the results that are not described in the methodology section. For instance, cumulative distribution functions (CDFs) are discussed in line 357, yet they are not explained in the methods. Including these details would improve transparency and ensure a more complete understanding of the approach used.

**R1.4:** We included the equation for the error formulas in a revised version of the Supplementary Information (see, Table S1). A brief explanation of cumulative distribution functions is added to Sect. 4.4 (lines 407:411):

*"These findings are reflected by cumulative distribution functions (CDFs) in Fig. 8.a and Fig. 8.b. The cumulative distributions shown in this study represent the change in the predictive accuracy of a model due to regional transfer. In other words, the CDFs provide insight into the extent to which a transferred flood loss model suffers from the different information contained in the Can Tho samples. The CDFs are shown in their normalized version to facilitate comparability despite different sample sizes."*

**C1.5:** In the introduction, the authors place emphasis on the case study to motivate the analysis. It may be beneficial to move the detailed description of the case study to a separate section, allowing the introduction to focus more directly on the research gaps. This would help to clearly establish the broader motivation and context for the study before delving into the specifics of the case study.

**R1.5**: Thank you for this suggestion. In the revised manuscript, we moved the description of the case studies to a new subsection (see, Sect. 2.1). Furthermore, we renamed Sect. 2 from "*Data – Post-flood survey of microbusinesses*" to "*Case studies and data*" (line 76) with subsections covering the case studies and the post-flood survey datasets. We hope this will make the broader motivation and context for the study clearer.

**C1.6:** The discussion and conclusion sections could be enhanced by further exploring how the findings may be utilized by other experts and their implications for flood risk management. Expanding on these aspects would clarify the broader relevance of the outputs.

**R1.6:** To illustrate the broader relevance of the study, we have improved the discussion and conclusions by highlighting the need of flood loss models to stakeholders and how reliable loss model predictions can improve decision making in terms of risk assessment. We added the following paragraph to Sect. 4.5 (lines 445:460):

*"Reliable flood loss models are essential tools for stakeholders and practitioners across multiple sectors, including insurance, urban planning, flood risk management, and climate adaptation decision-making. The flood loss models presented in this study specifically address the economic impacts of flooding on microbusinesses in Vietnamese cities, focusing on business interruptions and content losses. To our knowledge, this type of economic sector is underrepresented in risk management and the proposed models can advance decision making regarding this sector. By representing key drivers of loss as graph structures, the models offer an interpretable and transparent framework for understanding how various factors contribute to flood-related damages. The models are based on non-parametric Bayesian Networks, which enable probabilistic estimation of flood losses while explicitly quantifying uncertainty in both data and model formulation. This feature makes the models particularly robust, allowing for transparent assessment of risk and greater confidence in the results. Unlike traditional*

*deterministic models, the Bayesian approach ensures flexibility in handling incomplete or uncertain data, which is a common challenge in flood loss estimation. The combination of an interpretable model structure and transparent uncertainty quantification opens the door for operationalizing this modelling approach in practical settings. It provides stakeholders with a clearer understanding of how flood losses are calculated, promoting trust and facilitating decision-making. Furthermore, the model's ability to function effectively even with missing or limited data enhances its transferability to similar geographic regions and contexts. This adaptability is particularly valuable for expanding its application in data-scarce environments or in rapidly urbanizing areas where flood risks are evolving."*

Furthermore, we adapted and extended the following sentences in the conclusions (Sect. 5, lines 481:484):
*"The probabilistic flood loss models were calibrated and validated against reported flood losses in HCMC and in a transfer case study of Can Tho. The study resulted in interpretable and transferrable probabilistic flood loss models for predicting content losses and business interruption losses to microbusinesses."*

Below, we respond (**R**) to the more general and specific comments (**C**) of the reviewer. The changes in the revised text are mentioned in the response letter in *italics* along with line numbers. Furthermore, the revised manuscript includes one more figure (Fig. 1) and an additional subsection (Sect. 2.1).

General comments

**C2.0:** The manuscript addresses the topic of flood losses to microbusinesses in low- and middle-income countries and introduces an application of Conditional Random Forests for feature selection and Bayesian Networks to model flood losses, bridging gaps in existing methods that primarily cater to larger firms or macro-level analyses. In addition, it provides empirical evidence for key drivers of flood losses (e.g., water depth, building age, monthly revenue) and evaluates the models' transferability to another city (Can Tho), highlighting its potential for broader regional application. The approach is good in general; however, the manuscript should provide more details for readers to understand and reproduce this approach in other case studies.

**R2.0:** We thank the referee for taking the time and effort to provide comprehensive feedback on our manuscript. In the revised manuscript, we have implemented almost all the suggestions, otherwise, an explanation is provided in this response letter. Due to the length of the manuscript, we did not explain the individual pre-processing steps applied on the survey datasets in detail, but rather provided a brief overview, as mentioned in **R1.3** in the first response letter. We had also removed parts of the methodology section (e.g. the flowchart) in the manuscript to keep it as short as possible, however, we included these parts again in the revised versions of the manuscript and the Supplementary Information (e.g. flowchart in Fig. S1).

Specific comments

**C2.1:** Provide more explanation about machine learning techniques of "Bayesian Network" and "Conditional Random Forest" and briefly explain why these methods suit the study.

**R2.1:** Indeed, we have not explained the models in-depth due to the already rather large extent of the manuscript. As suggested, we have added brief explanations for both models and pointed out the suitability of the chosen methods compared to other ML-based approaches.

In the revised manuscript, we modified and extended following explanations of Conditional Random Forests in Sect. 3.1.1 (lines 189:197):
*"The candidate predictors for flood losses presented in Table 1 exhibit a moderate to high degree of multicollinearity, for instance, the flood-related features are strongly correlated to each other. For this reason, Conditional Inference Trees were applied to account for these correlations during feature selection. Conditional Inference Trees were initially introduced by Hothorn et al. (2006) and extended by Strobl et al. (2007) to an ensemble of trees, a so-called Conditional Inference Random Forest (CRF). Each tree is grown only by a subset of features, which were identified before as significant based on their p-values (Hothorn et al., 2006). By this approach predictive features are identified, despite their potential collinearity to other candidate predictors. The choice of an unbiased version of the permutation-based feature*

*importance method – namely Conditional Permutation Importance (CPI) – further reduces the chance of biased importance scores for correlated features (Debeer and Strobl, 2020)."*

In the revised manuscript, we added following explanations to Bayesian Networks (BNs) in Sect. 3.2.2 (lines 221:225):

*"They perform better in regional transfer settings compared to other ML-based models, such as regularized linear regressions, since BNs can be applied on incomplete information. Furthermore, they have the benefit of explicitly representing the dependency structures, quantifying uncertainty and the possibility of including expert knowledge alongside data. In more detail, the dependency structure of a BN represents (assumed) causal relations between variables, these dependencies can be set based on knowledge or logical conclusions."*

However, the manuscript does not deal with the conceptual basis of Bayesian Networks – the Bayes' theorem – or other aspects of them. For a more in-depth understanding, we would like to refer for CRF models to the work of Strobl et al. (2007) and Levshina (2020); for non-parametric Bayesian Network flood loss models to the work of Paprotny et al. (2021). In particular, the benefits of BNs briefly mentioned in the revised manuscript (lines 222-223) – "explicitly representing the dependency structures, quantifying uncertainty and the possibility of including expert knowledge alongside data" – are described in more detail in the latter study.

CRF:

> Levshina, N.: Conditional Inference Trees and Random Forests, in: A Practical Handbook of Corpus Linguistics, edited by: Paquot, M. and Gries, S. Th., Cham: Springer, 611–643, https://doi.org/10.1007/978-3-030-46216-1_25, 2020.

> Strobl, C., Boulesteix, A.-L., Zeileis, A., and Hothorn, T.: Bias in random forest variable importance measures: Illustrations, sources and a solution, BMC Bioinformatics, 8, 25, https://doi.org/10.1186/1471-2105-8-25, 2007.

BN:

> Paprotny, D., Kreibich, H., Morales-Nápoles, O., Wagenaar, D., Castellarin, A., Carisi, F., Bertin, X., Merz, B., and Schröter, K.: A probabilistic approach to estimating residential losses from different flood types, Nat. Hazards, 105, 2569–2601, https://doi.org/10.1007/s11069-020-04413-x, 2021.

**C2.2:** Case study selection: The manuscript justifies the choice of Ho Chi Minh City and Can Tho as case studies based on their high flood risk and economic importance. However, additional clarification is recommended (e.g., explain how these cities represent other flood-prone urban areas in Vietnam or Southeast Asia).

**R2.2:** This is a good point to further detail the choice of both cities in regard to their economic and topographic similarity to other flood-prone urban areas. The rapid economic developments in Southeast Asian states in the last decades are partly driven by the operations of micro-, small- and medium-sized businesses. The economic sector in urban areas of Vietnam, not only in HCMC and Can Tho, consists to a large share of those smaller businesses. The following sentence illustrates this aspect (Sect. 2.2.1, lines 128-129) "The presented shares of the business sectors in the HCMC survey are representative for entire Vietnam (General Statistics Office, 2018)" - HCMC is economically representative to other urban areas in Vietnam. Moreover, most of Vietnam's economic centres are located on flat terrain with access to the coast or a major river. Thus, they are similar flood-prone as HCMC and Can Tho, however, their type of flooding might differ (fluvial, pluvial, coastal). In the revised manuscript,

we adapted the following sentences to make the topographic similarity of HCMC to other urban areas more explicit (Sect. 2.1, lines 79-81):

"*Similar to other SE-Asian metropolises, HCMC lies in a river delta area close to the coast. These densely populated, flat, riverine and coastal regions experience regular flooding in particular during the rainy season (Garschagen, 2015; Tierolf et al., 2021; Nguyen et al., 2021).*"

**C2.3:** Defining sample size: The manuscript mentions the number of surveyed microbusinesses but lacks a detailed rationale for the sample size determination (What statistical considerations or sampling techniques were used to determine the sample size?).

**R2.3:** In fact, we did not adequately explain the different sample sizes in the manuscript and have therefore added the following sentences to Sect. 2.2.1 (lines 130:132):
"*In order to achieve a reasonable representation of HCMC, we selected the districts with the most frequent flood risk and heterogeneity in socio-economic conditions. Within each district, the shophouses were chosen randomly. The sample size in each district was not chosen based on statistical considerations, but on recommendation from local experts.*"

**C2.4:** Ethical application for surveys: Ethical considerations for conducting surveys need to be explicitly addressed to ensure transparency and compliance with research standards.

**R2.4:** Thank you for the very crucial comment. The survey resulted in anonymous data from flood affected businesses. We added following sentences to the beginning of Sect. 2.2 in the revised manuscript (lines 117-120):
"*They were informed about the project, how their responses would be used, and that they could leave the survey at any time. No personal or health-related information was collected in either survey. The data is stored and handled exclusively within the German Research Centre for Geosciences (GFZ) in compliance with data privacy and data protection regulations.*"

To connect the data of the interviews with flood characteristics, the location of the interviewed businesses was recorded. We decided to show in a GIS map (**R2.6**) only the approximate geolocations of the microbusinesses to protect the anonymity of the interviewees. The data stored and handled is in compliance with data privacy and data protection regulations.

**C2.5:** A flowchart summarizing the methodological workflow (from data collection to modeling and validation) can improve clarity for readers unfamiliar with machine learning or Bayesian methods.

**R2.5:** We moved the flowchart (Fig. S1) to the revised version of the Supplementary Information to not further extend the manuscript length. We linked the main steps, visualized by the flowchart, to the text in the data and methodology section.

**C2.6:** A GIS map that shows locations of surveyed microbusinesses can provide clear context and can be combined with flood hazard maps to give an overview of hazard and impact, which can be suitable for visualization and dissemination.

**R2.6**: We added a GIS map to the revised manuscript (Fig. 1) showing the approximate locations of the microbusinesses surveyed in HCMC. The entire urban area of HCMC as well as suburban districts are located on low-lying terrain, thus, in combination with many open channels, the majority of HCMC's districts are facing a high flood risk. We tried to illustrate this aspect in the map by adding information about the low-lying areas alongside the approximate locations of the microbusinesses. Besides the figure, we added the following text to the revised manuscript to explain this aspect (lines 143:145):

"*Figure 1 visualizes the approximate locations of the microbusinesses surveyed in HCMC. However, their exact geolocations are not shown to protect the anonymity of the interviewees. Furthermore, the map shows that the surveyed microbusinesses are often located near to an open channel or tributary river.*"

---

## Author Response (AR2)

**Response to Referee 1:**

Below, we respond (**R**) to the comment (**C**) of the reviewer.

**C1.1:** I have only a minor suggestion regarding the case study map, where slight improvements in cartographic design (e.g., font balance) could enhance clarity.

**R1.1:** We thank the referee for providing further feedback on the manuscript. As suggested, we decreased the font size of the title and increased the font size of the data sources used (Figure 1). In addition, the overall layout of the case study map was slightly rearranged.

**Response to Referee 2:**

Below, we respond (**R**) to the minor comments (**C**) of the reviewer. The changes in the revised text are mentioned in the response letter in *italics* along with line numbers.

**C2.1:** The abstract is concise and covers key aspects, but the mention of the mean absolute error for business interruption losses (18.7%) without context may confuse general readers. A brief explanation of its significance would be beneficial.

**R2.1:** We thank the referee for providing feedback and improved the clarity of the abstract by adapting the respective sentence (lines 20:22):
*"The models estimated the flood losses to HCMC's microbusinesses with a mean absolute error of 3.8 % for content losses (observed mean: 4.7 %) and 18.7 % for businesses interruption losses (observed mean: 18.2 %)."*

**C2.2:** In methodology, the rationale behind separating content loss modeling into "chance of loss" and "degree of loss" is clear, but a brief mention of why business interruption losses were not split in the same way would improve clarity.

**R2.2:** The manuscript pointed out that most interviewees in HCMC reported non-zero interruption losses, thus, the aspects of business interruption loss (chance and degree of interruption loss) were not considered separately (lines 183:185). We understand that the large bar of more than 140 small-loss cases in Figure S3 (Supplementary Information) could indeed be misinterpreted, but it includes besides cases of zero-loss also a large share of minor loss cases. To avoid misinterpretation of Figure S3, we added following sentence to the caption (lines 47:48, Supplementary Information):
*"The first bar on the left side of the HCMC dataset contains cases of zero-losses as well as a large share of cases representing minor losses (<5 % interruption losses)."*

**C2.3**: In results and discussion, the presentation of key drivers of flood losses is strong, but explaining the practical implications of each factor (e.g., how business revenue affects loss severity) could enhance its impact.

**R2.3**: We thank the referee for the suggestion. We adapted the explanation for business revenues in Section 4.2 (lines 314:319):
*"The revenue from business operations (**mthly. sales**) is positively correlated with the degree of content loss in the respective BN graph as shown in Fig. 3 (rho: 0.29), but only a weak positive correlation exists to relative interruption losses. Monthly sales are seen as an indicator for the microbusiness size and its type of business content, as they reflect the heterogeneity among companies (Schoppa et al., 2020). The level of sales affect both exposure and vulnerability. Higher sales can increase exposure by driving expansion into risk-prone areas and requiring larger inventories, which are more susceptible to extreme weather events."*

**C2.4:** The study's reliance on survey data is acknowledged, but more discussion on potential biases (e.g., underreporting of losses) would strengthen the arguments.

**R2.4:** Thank you for highlighting this crucial aspect. We extended Section 4.5 with a brief discussion of potential biases in the survey data as a limitation of the study (lines 464:469):

*"Despite these advantages, the models rely on empirical post-event survey datasets and have certain limitations. For instance, the sample was obtained voluntarily, which may introduce selection bias. The study focused on frequently flooded regions, including both well-established city areas and newly urbanized zones, to represent the city's expansion. However, the absence of official loss data prevents validation of the reported figures, particularly given the potential for under-reporting."*